# Four-dimensional hydrogel dressing adaptable to the urethral microenvironment for scarless urethral reconstruction

Yujie Hua[1,2,5], Kai Wang[3,5], Yingying Huo[2,5], Yaping Zhuang [ORCID][4], Yuhui Wang[1], Wenzhuo Fang[1], Yuyan Sun[2], Guangdong Zhou[2], Qiang Fu[1] ✉, Wenguo Cui [ORCID][4] ✉ & Kaile Zhang[1] ✉

The harsh urethral microenvironment (UME) after trauma severely hinders the current hydrogel-based urethral repair. In fact, four-dimensional (4D) consideration to mimic time-dependent physiological processes is essential for scarless urethral reconstruction, which requires balancing extracellular matrix (ECM) deposition and remodeling at different healing stages. In this study, we develop a UME-adaptable 4D hydrogel dressing to sequentially provide an early-vascularized microenvironment and later-antifibrogenic microenvironment for scarless urethral reconstruction. With the combination of dynamic boronic ester crosslinking and covalent photopolymerization, the resultant gelatin methacryloyl phenylboronic acid/*cis*-diol-crosslinked (**GMPD**) hydrogels exhibit mussel-mimetic viscoelasticity, satisfactory adhesion, and acid-reinforced stability, which can adapt to harsh UME. In addition, a temporally on-demand regulatory (**TOR**) technical platform is introduced into **GMPD** hydrogels to create a time-dependent 4D microenvironment. As a result, physiological urethral recovery is successfully mimicked by means of an early-vascularized microenvironment to promote wound healing by activating the vascular endothelial growth factor (VEGF) signaling pathway, as well as a later-antifibrogenic microenvironment to prevent hypertrophic scar formation by timing transforming growth factor-β (TGFβ) signaling pathway inhibition. Both in vitro molecular mechanisms of the physiological healing process and in vivo scarless urethral reconstruction in a rabbit model are effectively verified, providing a promising alternative for urethral injury treatment.

Urethral injury is a common and complicated disease in clinical practice that inevitably destroys multiple types of urethral architecture, such as the urothelium, blood vessels, and submucosal tissue[1–3]. Recently, three-dimensional (3D) hydrogels have attracted great attention as ideal wound dressings due to the advantages of wet-healing conditions, ease of carrying bioactive substances, and availability to mimic the cell microenvironment[4–8]. However, severe damage to the urethral microenvironment (UME) after trauma is an inherent obstacle that hinders the current hydrogel-based urethral repair[9–11]. In particular, robust adhesion to dynamic wound surfaces suffering from acidic urine in a harsh UME has proven to be highly challenging[12–15]. Among all types of adhesive hydrogels, boronic ester crosslinking between phenylboronic acid and *cis*-diol derivatives commonly exhibits mechanically dynamic self-healing and pH-dependent characteristics, as well as *cis*-diol-based adhesive properties analogous to those of catechol mussel adhesive protein[16–20]. In

addition, the combination of dynamic and covalent tandem cross-linking strategies could further enable the formation of mechanically stable hydrogels with desirable viscoelastic properties similar to those of the muscular urethra[21,22].

In addition to the above consideration of adaptability to the UME, ideal urethral reconstruction is a time-dependent physiological process instead of an autogenously pathological recovery, which commonly leads to serious urethral stricture due to hypertrophic scar formation[23,24]. As far as we know, the key point in achieving scarless urethral reconstruction is to balance extracellular matrix (ECM) deposition and remodeling at different healing stages. For example, it is necessary to build abundant vessel networks to transport blood and nutrients to support the survival of surrounding epithelial cells and fibroblasts at the early stage[25,26]. However, fibroblasts need to be reasonably harnessed to prevent hypertrophic scarring of obstructing the urethral lumen at the later stage[27,28]. To date, the current hydrogel-based treatments to efficiently promote wound healing against urethral stricture remain unreliable. Therefore, a better urethral repair strategy needs to transcend the traditional 3D hydrogel-based design and create a time-dependent four-dimensional (4D) microenvironment to fit well with different healing stages. Thus, it is necessary to effectively promote early-stage vascularized urothelium regeneration and simultaneously prevent later-stage excessive fibrogenesis.

Recent advances in hydrogel-based 4D cell culture platforms with spatiotemporal tunability represent a great opportunity to mimic the dynamic heterogeneity of the UME[29–32]. The hydrogel design of on-demand delivery of bioactive substances, such as biological growth factors or inhibitors, has emerged as the most efficient way to precisely regulate tissue regeneration[33–35]. Noticeably, the vascular endothelial growth factor (VEGF) signaling pathway plays a crucial role in the process of angiogenesis and epithelialization, which has been well recognized as a key regulator for promoting wound healing at the early stage of urethral reconstruction[36,37]. Dysregulated transforming growth factor-β (TGFβ) signaling pathway contributes heavily to pathological wound scarring, which exhibits an adverse effect on the formation of urethral strictures[38,39]. As previously reported, the timing of TGFβ signaling inhibition has been proven effective for scarless wound healing[40], which may provide a feasible approach to prevent hypertrophic scar formation at later stage of urethral reconstruction. Taken together, how to construct a time-dependent 4D modulation is essential to balance ECM deposition and remodeling for scarless urethral reconstruction.

Herein, we developed a UME-adaptable 4D hydrogel dressing to sequentially provide an early-vascularized microenvironment and later-antifibrogenic microenvironment for scarless urethral reconstruction. (Fig. 1). First, gelatin methacryloyl phenylboronic acid/*cis*-diol-crosslinked (**GMPD**) hydrogels were meticulously designed by a hybrid crosslinking strategy combining dynamic boronic ester crosslinking and covalent photopolymerization, which possessed mussel-mimetic viscoelasticity, satisfactory adhesion, and acid-reinforced stability adapted to the harsh UME. In addition, a temporally on-demand regulatory (**TOR**) technical platform, preferentially releasing VEGF through gelatin methacryloyl (**GM**) microgels and then releasing TGFβ inhibitor through poly-lactic-co-glycolic acid (**PLGA**) micro-capsules, was introduced into **GMPD** hydrogels to create a time-dependent 4D microenvironment. The molecular mechanism explorations revealed that **TOR**-functionalized **GMPD** hydrogel dressing could effectively provide an early-vascularized microenvironment to promote wound healing by activating the VEGF signaling pathway, as well as a later-antifibrogenic microenvironment to prevent hypertrophic scar formation by timing TGFβ signaling pathway inhibition. In contrast to pathological wound healing repaired by traditional 3D hydrogels, our UME-adaptable 4D hydrogel dressings could successfully achieve scarless urethral reconstruction in a rabbit model.

## Results

### Fabrication of UME-adaptable hydrogels

In our design, we first synthesized two types of gelatin-derived polymers: fluorophenylboronic acid (FPBA)-modified gelatin methacryloyl (**GMP**, ~32% FPBA-substituted degree; ~60% methacryloyl-substituted degree) and *cis*-diol-modified gelatin methacryloyl (**GMD**, ~39% diol-substituted degree; ~60% methacryloyl-substituted degree). To prepare UME-adaptable viscoelastic hydrogels, **GMPD**+*hv* hydrogels (mixing **GMP** and **GMD** polymers in an equivalent ratio) were synergistically crosslinked via both boronic ester dynamic bonds (viscous segment) and photoinitiated covalent bonds (elastic segment), whereas **GMPD**-*hv* hydrogels were only crosslinked via boronic ester dynamic bonds (Fig. 2A, B and Supplementary Fig. 1). Noticeably, **GMPD** hydrogels exhibited good shear-shinning and self-healing properties due to the inherent reversible crosslinked network by means of boronic ester dynamic bonds, which are conveniently injected onto the target site and completely cover the urethral defects (Supplementary Fig. 2).

Furthermore, [^1]H NMR spectra of **GMPD** hydrogels were performed to trace the crosslinking mechanism. As shown in Fig. 2C, both FPBA-compound characteristic peaks at 7.3–7.5 ppm and *cis*-diol-compound characteristic peaks at 3.1 ppm simultaneously appeared in **GMPD** hydrogels via boronic ester bond crosslinking. Moreover, the disappearance of double bond characteristic peaks at 5.2–5.7 ppm confirmed the further photopolymerization of methacryloyl groups. The hybrid crosslinking mechanism was also monitored by X-ray photoelectron spectroscopy (*XPS*) and attenuated total reflection Fourier transform infrared (ATR-FTIR) spectroscopy. As shown in Fig. 2D, the *XPS* results revealed that **GMPD**-*hv* hydrogels exhibited a moderate intensity of C-O (286.37 eV) species compared with **GMP** (lower intensity) and **GMD** (higher intensity) polymers, consistent with the fact that the **GMD** component contained more oxygen molecules in pedant alcohol groups. A new component associated with boronic ester bonds appeared clearly at 189.75 eV, close to the typical peak of boronic bonds at 191.34 eV (**GMP** component), which is attributable to successful boronic ester crosslinking (Fig. 2E). The further photopolymerization of **GMPD**+*hv* hydrogels has no significant effect on the molecular structure of the preformed dynamic boronic ester bonds according to both the C(1 s) and B(1 s) spectra (Supplementary Fig. 3). In addition, ATR-FTIR spectra of **GMPD**+*hv* hydrogels showed a definite decrease in the methacryloyl-related infrared band ratio of C-H stretching vibrations at 3100 cm[^-1] and C = C stretching vibrations at 1450 cm[^-1] after light irradiation (Fig. 2F), consistent with the [^1]H NMR spectra trace of the photopolymerization reaction. Taken together, these results essentially confirm the hybrid crosslinking mechanism of combining dynamic boronic ester crosslinking and covalent photopolymerization.

Rheological analyses were further conducted to investigate the viscoelastic properties of hybrid crosslinked hydrogels. First, frequency sweep rheological measurements were performed within the linear region to quantify the gel strength according to the crossover frequency ($\omega_c$) at which the storage modulus (G') was equal to the viscous modulus (G"). As shown in Fig. 2G, **GMPD**-*hv* hydrogels exhibited frequency-dependent viscoelastic behavior, a typical feature of dynamic gel networks via boronic ester bonds. However, **GMPD**+*hv* hydrogels showed more elastic performance and higher mechanical strength than **GMPD**-*hv* hydrogels due to the further enhancement of gel networks via photopolymerization. Importantly, the dynamic crosslinking of boronic ester bonds exhibited acid-reinforced mechanical strength with $\omega_c$ increasing as pH decreased (ranging from 4.5 to 8.5), which is very suitable for harsh UME of low pH values (Fig. 2H, I). Moreover, time sweep rheological measurements demonstrated that secondary photopolymerization could effectively improve the elasticity of **GMPD** hydrogels without any influence of pH values (Fig. 2J). As a result, the hybrid

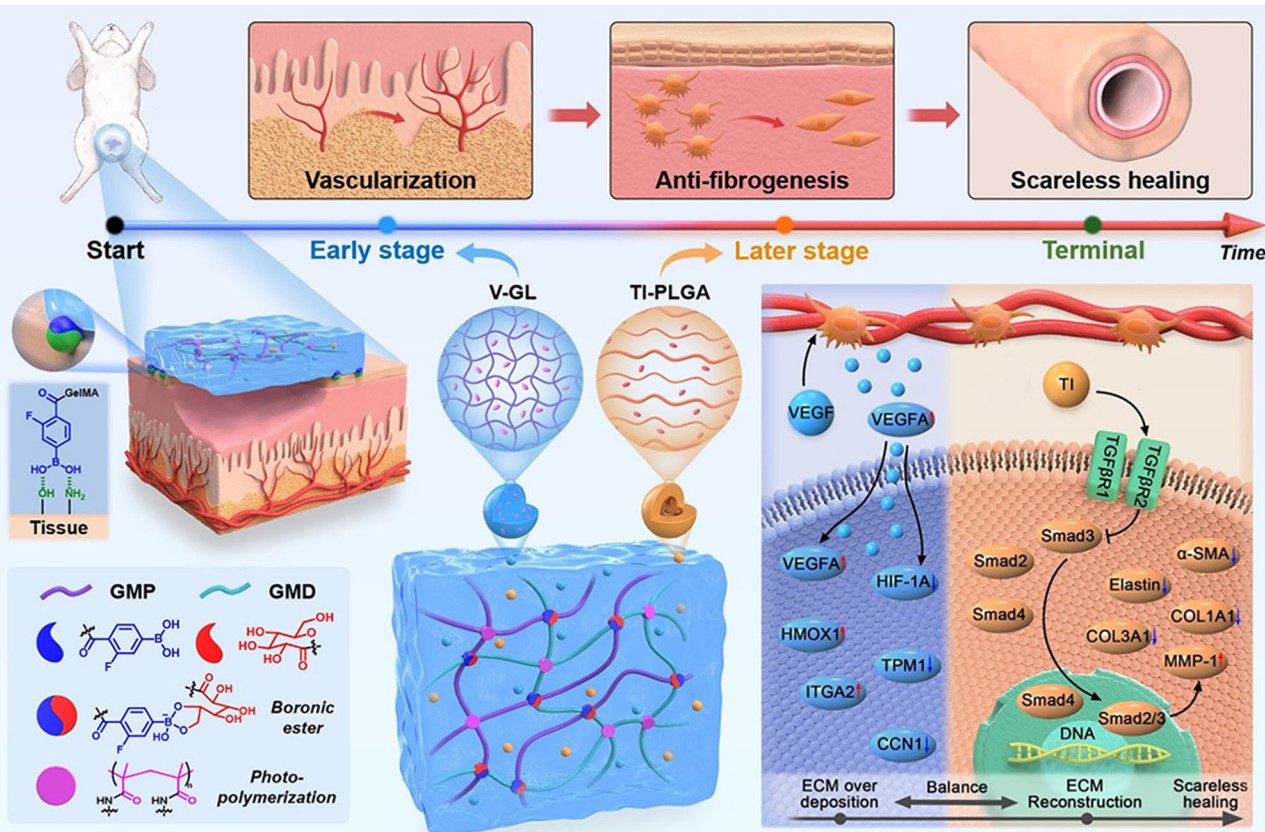

**Fig. 1 | The design concept of 4D hydrogels.** Schematic illustration of UME-adaptable 4D hydrogel dressing to fully mimic time-dependent scarless urethral reconstruction.

crosslinking strategy not only provides viscous segments relying on boronic ester dynamic bonds but also offers elastic segment dependent on photoinitiated covalent bonds, and thus achieves satisfactory viscoelasticity that can closely mimic the tissue architecture of urethral mussels.

**Performance of tissue adhesion and acid-reinforced stability**

As previously reported, phenylboronic acid has a "*cis*-diol" moiety that exhibits adhesive properties similar to those of catechol mussel adhesive protein[17,18]. In this study, **GMPD** hydrogels had the potential to balance hydrogel reversibility and adhesion strength based on the hybrid crosslinking strategy (Fig. 3A). To evaluate the tissue binding ability, **GMPD** hydrogels were gelled in situ on the surface of muscle upon light irradiation and then immersed in acid solution to sustain stability. Figure 3B depicted that there was no obvious breakage or detachment of the adhesive hydrogels regardless of stretching and twisting behavior upon acidic solution (pH = 4.5). The scanning electron microscopy (SEM) results clearly revealed a tight and seamless interface between **GMP** hydrogels and muscle upon photopolymerization, whereas **GM** hydrogels showed obvious interfacial separation against the surrounding muscle, which is consistent with the structural predictions between the "*cis*-diol" moiety of **GMP** hydrogels and active hydrogen on the tissue surface (Fig. 3C). To quantitatively evaluate the adhesion performance of **GMPD** hydrogels as wound dressings, standard lap shear and incision sealing strength were measured respectively. As shown in Fig. 3D–I, the peak adhesive strength (*l*) and sealing strength (*i*) of **GMPD-*hv*** hydrogels (*l* = 19.4 ± 2.5 kPa; *i* = 3.1 ± 0.6 N) were higher than those of **GM** hydrogels (*l* = 1.1 ± 0.2 kPa; *i* = 0.6 ± 0.1 N) and commercially available fibrin glue (*l* = 12.2 ± 1.7 kPa; *i* = 1.4 ± 0.3 N), which was probably attributed to the *cis*-diol-mediated adhesive capability. Noticeably, the tensile

strength required to dislocate hydrogels from tissue significantly increased as secondary covalent stabilization via photopolymerization (**GMPD+*hv*** hydrogels, *l* = 47.7 ± 4.3 kPa; *i* = 9.6 ± 1.2 N), indicating that the enhancement of mechanical stability could effectively improve the adhesive strength. The ex vivo adhesive performance was basically unchanged after 12 hours, which verified the stability of hydrogel-tissue adhesion (Supplementary Fig. 4). More importantly, the **GMPD +*hv*** hydrogels could also withstand the flow-induced shear force in vivo, which is essential for adapting to the wet and dynamic urethral environment (Supplementary Fig. 5). Gel stability is another important parameter for wound dressings, especially in dynamic and acidic urethral environment. As shown in Supplementary Fig. 6, **GMPD-*hv*** hydrogels in acid solution could effectively extend the in vitro degradation time from ~3 days (pH = 7.4) to ~14 days (pH = 4.5), which is attributed to the acid-reinforced structural stability of boronic ester bonds (correlated with Fig. 2I). In addition, **GMPD+*hv*** hydrogels exhibited long-lasting stability over 14 days in any harsh UME (pH values from 4.5 to 7.4) due to the pH-independent covalent stabilization of photopolymerization. The in vivo degradation experiments further showed the remaining hydrogels fully covered the urethral defects after 14 days, which could basically meet the requirement of wound healing process (Supplementary Fig. 7). In addition, **GMPD+*hv*** hydrogels didn't show obvious swelling ratio in acidic pH value due to the secondary covalent photopolymerization (Supplementary Fig. 8). Therefore, **GMPD** hydrogels are suitable for dynamic and acidic urethral environment based on the characteristics of acid-reinforced and covalent-stabilized mechanical properties. All these results demonstrated that our **GMPD** hydrogels possessed facile operation, mussel-mimetic viscoelasticity, satisfactory adhesion, and acid-reinforced stability, which could serve as an ideal wound dressing applied in the harsh UME.

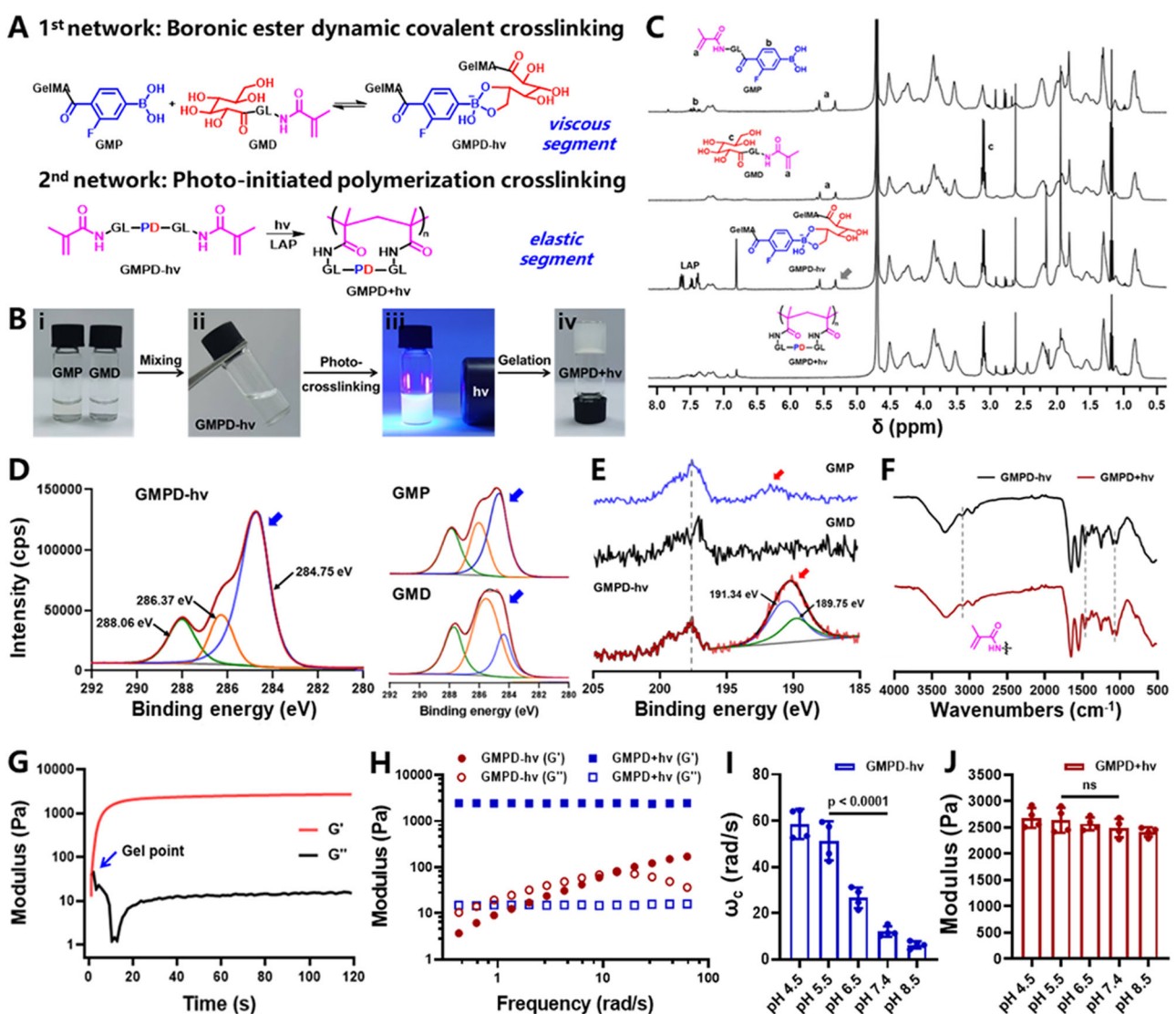

**Fig. 2 | Physicochemical characterization of UME-adaptable GMPD hydrogels.**
**A** Schematic of hybrid crosslinking mechanisms obtained by combining boronic ester dynamic crosslinking (viscous segment) and photopolymerization (elastic segment) to construct viscoelastic **GMPD** hydrogels. **B** Photographs show gelation steps through two-component mixing followed by light irradiation. **C** ¹H NMR spectra evolution of **GMPD** hydrogels with or without light irradiation. Gray arrows represent the proton peak of methacryloyl groups. **D** The C(1 s) *XPS* regions of **GMP, GMD**, and **GMPD-*hv*** hydrogel samples. Intensity (cps): count per second. Red line: C(1 s) characteristic speak; green line: C = O (288.06 eV); orange line: C-O (286.37 eV); blue line: C-C (284.75 eV); blue arrows represent C(1 s) regions. **E** The B(1 s) XPS regions of **GMP, GMD**, and **GMPD-*hv*** hydrogel samples. Red line: B(1 s) characteristic speak; blue line: B-O (191.34 eV); green line: B-C (189.75 eV); red

arrows represent B(1 s) regions. **F** ATR-FTIR spectra of **GMPD** hydrogels with or without light irradiation. **G** Representative time sweep rheological plots of **GMPD** +***hv*** hydrogels. **H** Representative frequency sweep rheological plots of **GMPD-*hv*** and **GMPD+*hv*** hydrogels at pH 7.4. **I** The crossover frequencies (ω$_c$) obtained from frequency sweep rheology for **GMPD-*hv*** hydrogels at each pH value ranging from 4.5 to 8.5. **J** Final shear moduli of **GMPD+*hv*** hydrogels at each pH value ranging from 4.5-8.5. **GMPD** hydrogels: 10% w/v, **GMP: GMD** = 1:1; **GMPD-*hv*** hydrogels: **GMPD** hydrogels without light irradiation; **GMPD+*hv*** hydrogels: **GMPD** hydrogels with light irradiation; light: 365-nm LED, 20 mW/cm², 30 s irradiation time. *n* = 4 independent samples. Data are presented as mean ± SD. All error bars represent SD. *p* values calculated using one-tailed unpaired *t*-test. ns = no significance. Source data are provided as a Source Data file.

## In vitro biological evaluation of the TOR platform

To better mimic the dynamic physiological process of urethral reconstruction, it is necessary to efficiently promote wound healing at the early stage and prevent hypertrophic scarring at the later stage. As shown in Fig. 4A, a **TOR** technical platform was developed to address the above challenge by preferentially releasing VEGF from **V-GM** microgels (~58.7% encapsulation efficiency) and subsequently releasing a TGFβ inhibitor (SB431542) from **TI-PLGA** microcapsules (~83.6% encapsulation efficiency). The microscope and SEM examinations demonstrated that microfluidic-based **GM** microgels showed uniform transparent spherical morphology and the lyophilized samples displayed a typical porous structure on the microgel surface (Fig. 4B). In addition, **PLGA** microcapsules prepared by the

water-oil-water (W/O/W) double emulsion strategy possessed a suitable shell thickness to suppress the initial burst release and a hollow inner structure to encapsulate insoluble drugs (Fig. 4C). The drug release kinetics were further tested to investigate the feasibility of the temporally on-demand regulatory strategy. As shown in Fig. 4D, **V-GM** microgels exhibited initial burst release within ~3 days, meanwhile the core-shell structure of **TI-PLGA** microcapsules could effectively delay the release of the TGFβ inhibitor after 3 days and sustain lasting release until 10 days, which basically conformed to the time-dependent process of urethral reconstruction as previously reported[41,42]. Together, the **TOR** platform was successfully fabricated by means of early-released **V-GM** microgels and later-released **TI-PLGA** microcapsules.

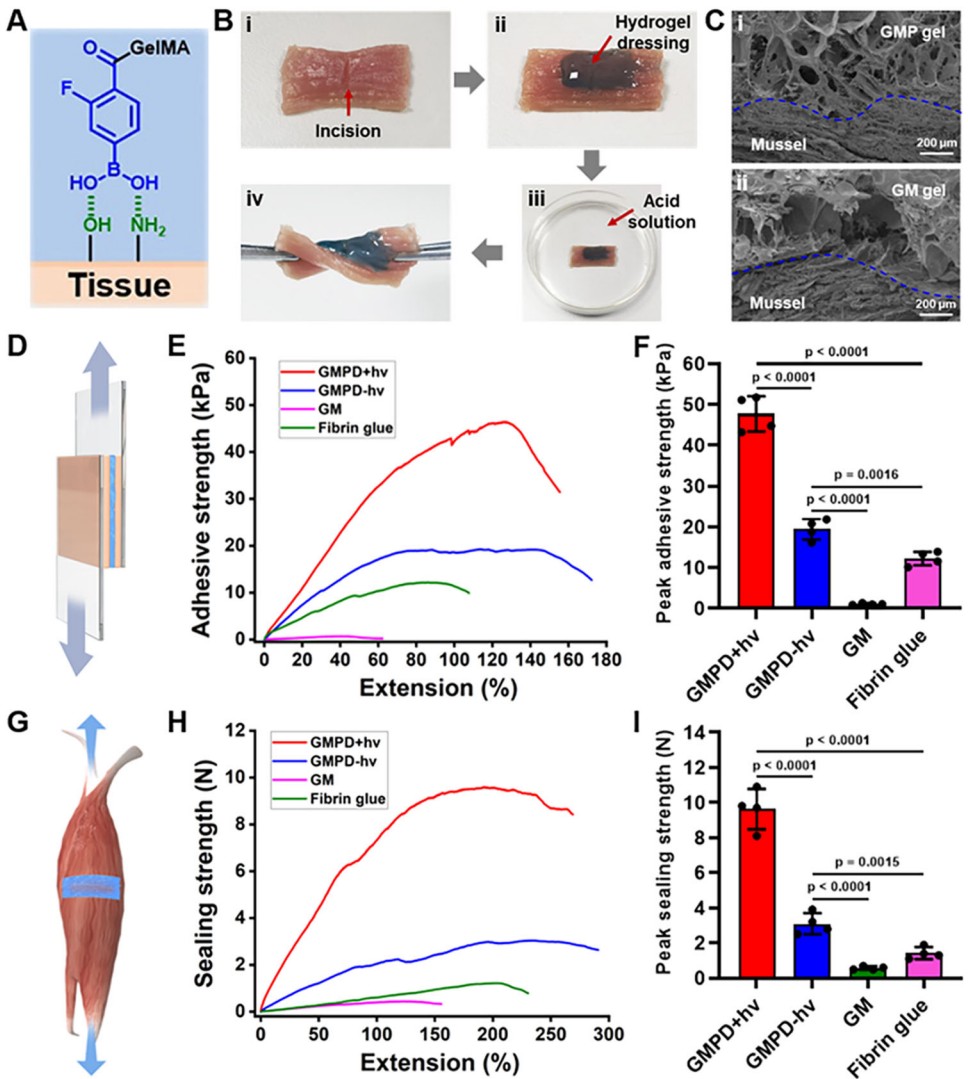

**Fig. 3 | Adhesion performance of GMPD hydrogels. A** Schematic of *cis*-diol-based adhesive mechanism via **GMP** component. **B** Photographs of **GMPD** hydrogels (stained by fast green) formed in situ on muscle tissue and immersed in acidic solution (pH = 4.5). **C** Representative SEM images of adhesive **GMP** hydrogel-mussel constructs compared with those of nonadhesive **GM** hydrogels (*n* = 3 independent samples). **D–F** Standard lap shear tests to determine the hydrogel-tissue binding strength of **GMPD+*hv*** hydrogels compared with those of **GM,** **GMPD-*hv*,** and fibrin glue. **G–I** Standard incision sealing tests to measure the hydrogel-mussel binding strength of **GMPD+*hv*** hydrogels compared with those of **GM, GMPD-*hv*,** and fibrin glue. **GMP** hydrogels: 10% w/v FPBA-modified GelMA; **GM** hydrogels: 10% w/v GelMA. The other hydrogel compositions are the same as in Fig. 2. *n* = 4 independent samples. Data are presented as mean ± SD. All error bars represent SD. *p* values calculated using one-tailed unpaired *t*-test. Source data are provided as a Source Data file.

Furthermore, the biological function of the **TOR** platform was evaluated to verify the time-dependent wound healing process. As shown in Fig. 4E–G, cell scratch experiments demonstrated that fibroblasts in the **V-GM** groups reached nearly 100% coverage at 24 h and showed the highest proliferation rate due to the initial release of VEGF, whereas the **TI-PLGA** groups reached ~75% coverage at 24 h and exhibited the suppression of cell proliferation on day 7 attributed to the gradual release of the TGFβ inhibitor. In addition, HUVECs in the **V-GM** groups showed an accelerated healing rate, and the **TI-PLGA** groups showed no significant difference in HUVEC proliferation even with the release of the TGFβ inhibitor (Supplementary Fig. 9). The same results were further confirmed by both fibroblast- and HUVEC-specific fluorescence staining. As shown in Fig. 4H–L and Supplementary Fig. 10, fibroblast-specific expression (fibronectin, α-SMA, COL1, and COL3) was significantly upregulated in the **V-GM** groups, whereas the corresponding expression were dramatically down-regulated in the **TI-PLGA** groups, implying successful inhibition of fibroblast-specific function owing to the gradual release of the TGFβ

inhibitor. The HUVEC-specific fluorescence staining (CD31 and VWF) showed that the initial release of VEGF effectively upregulated vascularized-related expression (Supplementary Fig. 11). In short, the meticulously designed **TOR** platform has a synergistic effect of promoting vascularization for wound healing at the early stage and preventing fibrogenesis at the later stage, and thus provides a 4D cell culture system combining a time-dependent **TOR** platform with UME-adaptable hydrogels.

## Scarless healing molecular mechanism of the time-dependent process

Before investigating the 4D dynamic regulation of the scarless healing process, the cytocompatibility was evaluated using CCK-8 assays. As shown in Supplementary Fig. 12, both **GMPD** hydrogel and **TOR** microsphere extracts did not show obvious cytotoxicity (>92% cell viability) for either fibroblasts or HUVECs. To exploit the molecular mechanism involved in scarless wound healing based on **TOR**-functionalized hydrogel dressings, differentially expressed genes (DEGs) were

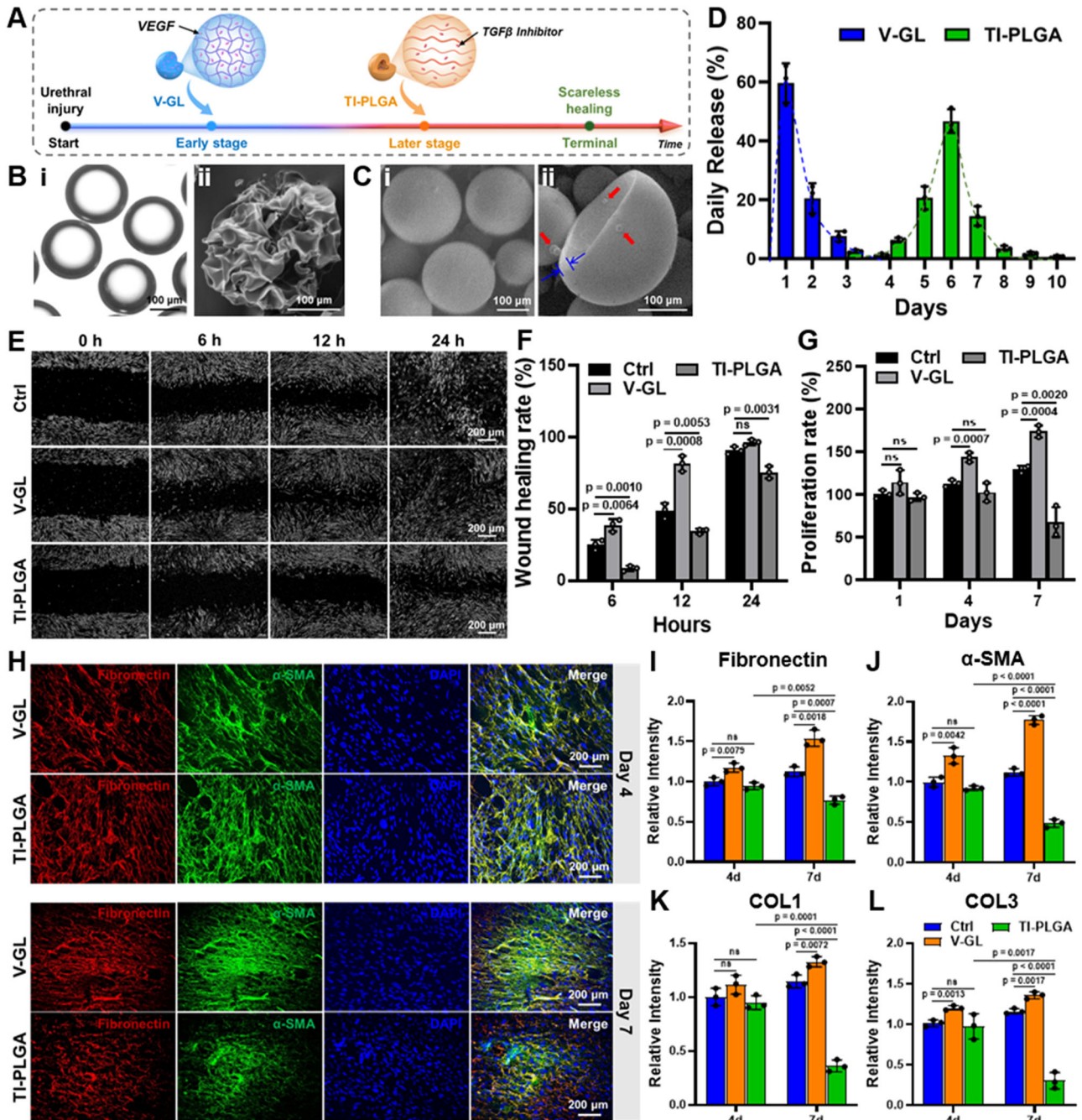

**Fig. 4 | In vitro biological evaluation of the TOR platform. A** Schematic of the design of the **TOR** platform by means of early-released **V-GM** microgels and later-released **TI-PLGA** microcapsules. **B** Representative microscope (i) and SEM (ii) images of **V-GM** microgels (*n* = 3 independent samples). **C** Representative SEM images of the intact (i) and crushed (ii) **TI-PLGA** microcapsules. Blue arrows represent the shell of **PLGA** microcapsules. Red arrows represent the cargo of TGFβ inhibitor powders (*n* = 3 independent samples). **D** The daily drug release curves of **V-GM** microgels and **TI-PLGA** microcapsules in **GMPD+***hv* hydrogels (*n* = 3 independent samples). **E** Observation of the fibroblast scratch assay after 0, 6, 12, and 24 h in the **V-GM, TI-PLGA**, and control (**Ctrl**) groups. **F** Statistical analyses of the corresponding wound healing rates. **G** Cell proliferation rates of fibroblasts tested by CCK-8 assay after 1-, 4-, and 7-days culture in the **V-GM, TI-PLGA**, and **Ctrl** groups. **H** Immunofluorescence staining of fibronectin (red), α-SMA (green), and cell nuclei (DAPI, blue) for fibroblast marker expression after 4- and 7-days culture in the **V-GM** and **TI-PLGA** groups. **I–L** Comparative fibrogenic expression levels (*fibronectin*, *α-SMA*, *COL1*, and *COL3*) after 4- and 7-days culture in the **V-GM, TI-PLGA**, and **Ctrl** groups. **TOR**: temporally on-demand regulatory; **TI-PLGA**: TGFβ inhibitor-loaded PLGA microcapsules; **V-GM**: VEGF-loaded GelMA microgels. *n* = 3 biologically independent samples. Data are presented as mean ± SD. All error bars represent SD. *p* values calculated using one-tailed unpaired *t*-test. ns = no significance. Source data are provided as a Source Data file.

identified and biological processes were investigated using RNA-seq methods (see Methods). The Venn diagram showed the total changes and overlaps of fibroblast DEGs in **TOR**-functionalized hydrogels after 1-, 4-, and 7-days culture (Fig. 5A). As shown in the volcano plots, there were 928 upregulated and 695 downregulated DEGs between the F1 and F4 groups at the early stage, as well as 267 upregulated and 382 downregulated DEGs between the F4 and F7 groups at the later stage (Fig. 5B). Moreover, the volcano plots of endothelial cell DEGs showed 1989 upregulated and 2277 downregulated genes between the E1 and E4 groups at the early stage (Supplementary Fig. 13A, B).

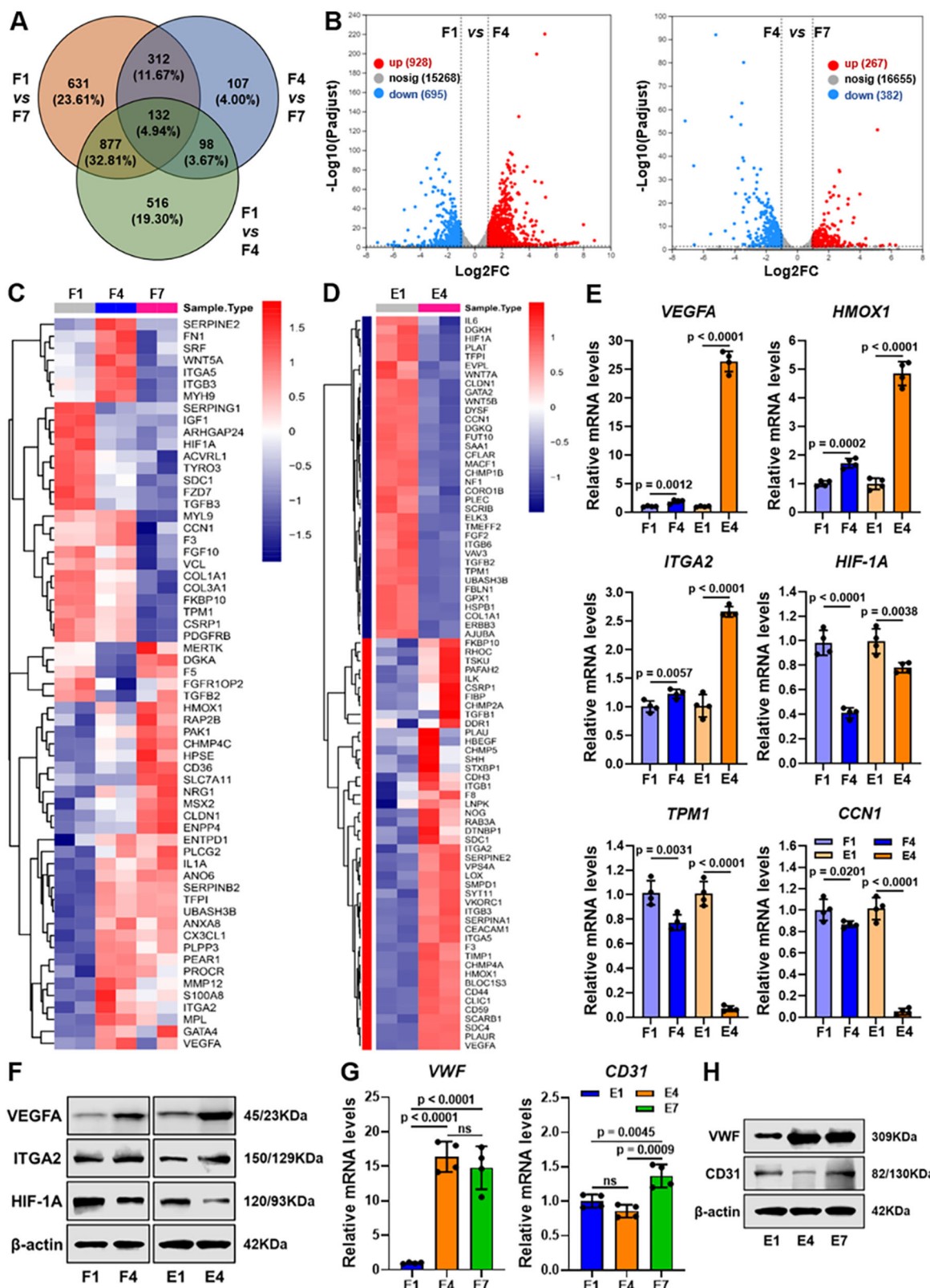

Next, DEGs of both fibroblasts and endothelial cells involved in the wound healing process at the early stage were determined to investigate the synergetic effect of the **TOR** platform. A total of 61 DEGs involved in the wound healing process at the early stage were dysregulated in fibroblasts (Fig. 5C), while 82 DEGs were dysregulated in endothelial cells at day 4 (Fig. 5D). Notably, three increased DEGs (*VEGFA*, *HMOX1*, and *ITGA2*) and four decreased DEGs (*HIF-1A*, *TPM1*, *CCN1*, and *COL1A1*) involved in the wound healing process at the early stage were commonly dysregulated in both fibroblasts and endothelial cells, which are very relevant to VEGF singling pathways. The corresponding DEGs were further validated by quantitative real-time polymerase chain reaction (qRT-PCR) and western blot (WB). As shown in Fig. 5E, gene expression (*VEGFA*, *HMOX1*, and *ITGA2*) was dramatically upregulated in the E4 groups compared to the E1 groups, and the same

**Fig. 5 | Mechanistic analysis of the synergetic effect on vascularized urethral regeneration at the early stage. A** Venn diagram of fibroblast DEGs in **TOR**-functionalized hydrogels after 1-, 4-, and 7-days culture. **B** The corresponding volcano plots analyzed between the F1 and F4 groups at the early stage, as well as between the F4 and F7 groups at the later stage. **C, D** Heatmaps of screened DEGs involved in the wound healing process of fibroblasts (**C**) and endothelial cells (**D**) in **TOR**-functionalized hydrogels. **E** Comparative gene expression (*VEGFA, HMOX1, ITGA2, HIF-1A, TPM1,* and *CCN1*) of fibroblasts and endothelial cells after 1- and 4-days culture. **F** The protein expression (VEGFA, ITGA2, and HIF-1A) of fibroblasts

and endothelial cells after 1- and 4-days culture. **G, H** The corresponding gene (**G**) and protein (**H**) expression levels (vWF and CD31) of endothelial cells in **TOR**-functionalized hydrogels after 1-, 4-, 7-days culture. DEGs: differentially expressed genes. F1, F4, and F7: fibroblasts after 1-, 4-, and 7-days culture; E1, E4, and E7: endothelial cells after 1-, 4-, and 7-days culture. *n* = 4 biologically independent samples. Data are presented as mean ± SD. All error bars represent SD. *p* values calculated using one-tailed unpaired *t*-test. ns = no significance. Source data are provided as a Source Data file.

gene expression was also significantly upregulated in the F4 groups compared to the F1 groups, indicating the synergetic promotion of angiogenesis associated with fibrogenesis formation. Similarly, the gene expression levels (*HIF-1A, TPM1,* and *CCN1*) were greatly down-regulated in both the E4 and F4 groups compared to the corresponding E1 and F1 groups. The same results for protein expression levels were also confirmed by WB experiments, as shown in Fig. 5F. In addition, the early index of the vascularized expression level (VWF) was dramatically upregulated between the E4 and E1 groups, and the later index of the vascularized expression level (CD31) were significantly upregulated between the E7 and E4 groups, demonstrating the activation of VEGF singling pathways via **V-GM** microgels regardless of the later release of the TGFβ inhibitor (Fig. 5G, H).

The singling pathways involved in the wound healing process were further investigated between the F1 and F4 groups at the early stage, as well as between the F4 and F7 groups at the later stage. Figure 6A depicted the top 15 potential biological processes and signaling pathways at the early stage, according to the Gene Ontology (GO) and Kyoto Encyclopedia of Genes and Genomes (KEGG) enrichment analyses, which were related to the terms of cell cycle, cellular component organization, and metabolic process. Notably, DEGs between the E1 and E4 groups at the early stage of the cell cycle, organelle, and cellular component organization processes were commonly enriched in GO and KEGG terms of endothelial cells, revealing that these processes are critical to wound recovery (Supplementary Fig. 13C, D). At the later stage of wound healing, cell adhesion, differentiation, extracellular matrix, and structure organization, as well as tissue development and blood vessel development, were dominant, implying that ECM reconstruction is an important event during wound recovery (Fig. 6B). Noticeably, enriched TGFβ signaling pathways were also observed by KEGG pathway analysis, which indirectly confirmed the essential regulatory roles of later-released TGFβ inhibitors in the wound healing process. As shown in Fig. 6C, D, qRT-PCR and WB experiments demonstrated that the fibrogenic expression levels (α-SMA, elastin, COL1, COL3, and fibronectin) were significantly down-regulated, which is vital for the matrix remodeling of excessive ECM at the later stage. The main explanations for the regulatory function of ECM reconstruction are attributable to the negative regulation of TGFβ2/Smad3 signaling (*TGFB2* and *Smad3* downregulated) associated with the activation of MMP-related matrix degradation (*MMP1* upregulated) upon the later-released TGFβ inhibitor through **TI-PLGA** microcapsules.

### In vivo scarless urethral reconstruction in rabbits
The in vivo biological effect of the **TOR**-functionalized hydrogel dressings was further investigated in rabbit models. As shown in Fig. 7A, viscoelastic hydrogel precursors were first injected to fully cover the urethral defects, followed by light irradiation (365-nm LED, 20 mW/cm²) for stabilization. The recovery of injured urethras in each group was detected after 4- and 8-weeks surgery. In the ultrasound urethrograms, both the **Ctrl** and **GMPD** groups displayed a very narrow lumen caused by excessive scar formation (Fig. 7B, C). The **GMPD-V** and **GMPD-TI** groups showed relatively wide urethral lumens to various extent, whereas the **GMPD-V/TI** groups exhibited a fluent lumen similar to that of a normal urethra. Gross morphology

demonstrated that urethral repair in the **GMPD-V/TI** groups was significantly better than that in the other groups, as the repaired urethras were smooth without any hypertrophic scar formation. However, both the **Ctrl** and **GMPD** groups exhibited undesired shrinkage of urethras, showing the worst repair, while excessive scar formation could be seen in the **GMPD-V** groups (Fig. 7D).

Histological analyses of urethral tissue at 8 weeks after surgery were performed by hematoxylin and eosin (H&E) and Masson's tri-chrome staining. The urethral condition of ECM deposition, urothelium regeneration, and tissue fibrogenesis was shown in Fig. 7E and Supplementary Figs. 14, 15. The repaired urethras in both the **Ctrl** and **GMPD** groups exhibited severe urethral stricture with incomplete urothelium layers, fewer blood vessels, and hypertrophic scar formation, which is mainly attributed to the lack of regenerated urothelium layers. In contrast, the repaired urethras treated by **GMPD-V** hydrogel dressings had almost normal urothelium layers with abundant blood vessels, but there was excessive collagen deposition under the regenerated epithelium. In the **GMPD-TI** groups, although there was no obvious ECM over deposition, the urothelium layers did not regenerate well. With the synergetic release of both VEGF and TGFβ inhibitor, the continued and complete urothelium layers were satisfactorily regenerated on the lumen surface without any hypertrophic scar formation in the **GMPD-V/TI** groups after 8-weeks surgery, which is regarded as scarless urethral reconstruction.

Immunofluorescence examinations were used to further evaluate protein expression in the repaired urethras (Fig. 8A, B and Supplementary Fig. 16). The expression level of epithelial cytokeratin AE1/AE3, an important membrane surface protein marker, was significantly higher in the **GMPD-V/TI** groups than in the other groups. Specifically, the regenerated urothelium layer in the **GMPD-V** and **GMPD-TI** groups was thinner than that in the **GMPD-V/TI** group, whereas the expression of cytokeratin in both the **Ctrl** and **GMPD** groups was hardly found. Noticeably, the **GMPD-V** and **GMPD-V/TI** groups showed large numbers of CD31-positive cells (blood vessels labeled by CD31), which were much more abundant than those in other groups, implying that the regulation of the VEGF signaling pathway has a significant role in improving angiogenesis under the epithelium. The α-SMA expression level (the marker of myofibroblasts) under the epithelium was relatively lower in the **GMPD-TI** and **GMPD-V/TI** groups than in the other groups, implying that the effective inhibition of the TGFβ signaling pathway is conducive to antifibrogenic function. However, both the **Ctrl** and **GMPD** groups showed highly expressed α-SMA protein, indicating that large numbers of myofibroblasts existed under the epithelium. Consistent with the results of α-SMA, the expression level of collagen I at the site of repaired urethras treated by **GMPD-V/TI** hydrogel dressing was relatively lower than that in other groups. In addition, the positive expression of proliferating cell nuclear antigen (PCNA) was obviously increased at the site of regenerated epithelium in the **GMPD-V/TI** groups. Therefore, it is suggested that the synergetic release of both VEGF and TGFβ inhibitors has the potential to promote the regeneration of urothelium layers without hypertrophic scar formation. Different degrees of inflammatory responses at the injured sites appeared due to the application of the extra hydrogel dressings (Supplementary Fig. 17). Noticeably, the current results demonstrated that the number of CD206-positive cells (M2

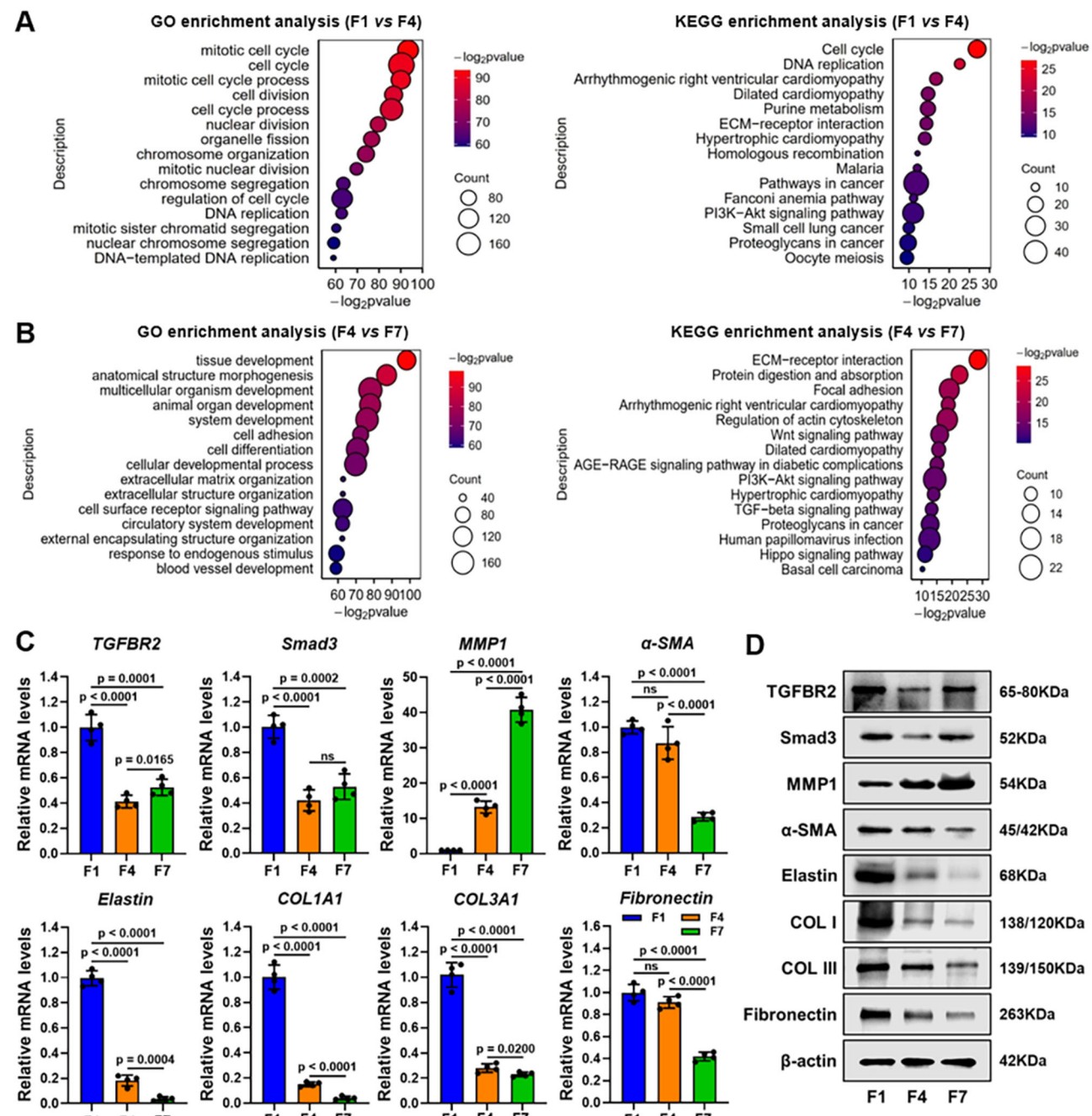

**Fig. 6 | Mechanistic analysis of antifibrogenic function for scarless urethral remodeling at the later stage. A**, **B** Gene Ontology (GO) and Kyoto Encyclopedia of Genes and Genomes (KEGG) enrichment analyses of fibroblast DEGs after mRNA sequencing in **TOR**-functionalized hydrogels between the F1 and F4 groups at the early stage (**A**), as well as between the F4 and F7 groups at the later stage (**B**), including the top 15 representative upregulated or downregulated signaling pathways. **C**, **D** The corresponding gene (**C**) and protein (**D**) expression levels (TGFBR2, Smad3, MMP1, α-SMA, elastin, COL1, COL3, and fibronectin) of fibroblasts in **TOR**-functionalized hydrogels after 1-, 4-, and 7-days culture, revealing the molecular mechanism of scarless urethral remodeling through inhibition of the TGFβ signaling pathway. $n = 4$ biologically independent samples. Data are presented as mean ± SD. All error bars represent SD. $p$ values calculated using one-tailed unpaired $t$-test. ns = no significance. Source data are provided as a Source Data file.

macrophages labeled by CD206) was highly expressed in both **GMPD-V** and **GMPD-V/TI** groups than those of other groups, which were conducive to urethral defect repair because M2 macrophages could effectively inhibit the inflammatory response and promote tissue regeneration.

All these results suggested that **TOR**-functionalized hydrogel dressings not only promoted the regeneration of vascularized urothelium layers via early-released VEGF but also effectively inhibited hypertrophic scar formation via later-released TGFβ inhibitor.

## Discussion

Urethral injury is a common disease that is usually accompanied by severe urethral stricture due to hypertrophic scar formation. In the clinic, free autogenous lingual mucosa tissue has been widely applied in upper urinary tract repair for more than ten years, but it is limited by the available sampling size, as well as adverse impacts on patients' pronunciation, mastication and other functions[43,44]. Recently, the development of tissue engineering methods that combine biocompatible scaffolds with adult cells or stem cells has effectively improved urethral

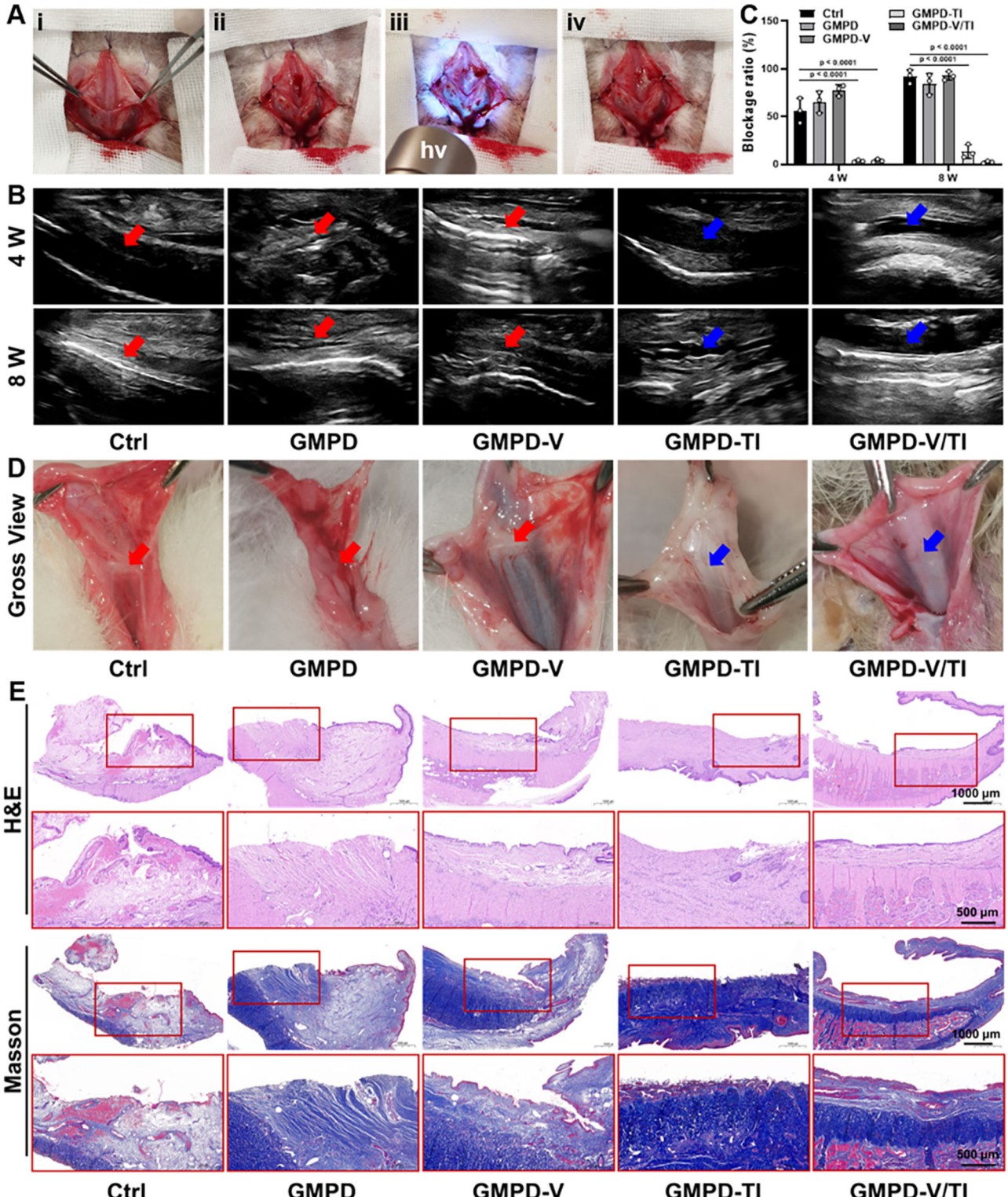

**Fig. 7 | In vivo scarless urethral reconstruction using GMPD hydrogel dressings in rabbits. A** Photographs of the surgical operation of in situ urethral defect repair using **GMPD** hydrogel dressings. **B**, **C** Urethrography images (**B**) and the corresponding blockage ratios (**C**) of the rabbit urethral canal in the **GMPD, GMPD-V, GMPD-TI, GMPD-V/TI**, and control (**Ctrl**) groups after 4- and 8-weeks surgery. Red arrows represent urethral stricture. Blue arrows represent urethral patency. **D**, **E** Gross view (**D**) and histological examinations of H&E and Masson's trichrome staining (**E**) of the rabbit urethral canal in the **GMPD, GMPD-V, GMPD-**TI, GMPD-V/TI, and control (**Ctrl**) groups after 8-weeks surgery. **GMPD** hydrogels: 10% w/v, **GMP: GMD** = 1:1; **GMPD-V** hydrogels: 10% w/v **GMPD** with 1% w/v **V-GM**; **GMPD-TI** hydrogels: 10% w/v **GMPD** with 1% w/v **TI-PLGA**; **GMPD-V/TI** hydrogels: 10% w/v **GMPD** with 1% w/v **V-GM** and 1% w/v **TI-PLGA**; light: 365-nm LED, 20 mW/cm$^2$, 30 s irradiation time. $n = 3$ biologically independent samples. Data are presented as mean ± SD. All error bars represent SD. $p$ values calculated using one-tailed unpaired $t$-test. Source data are provided as a Source Data file.

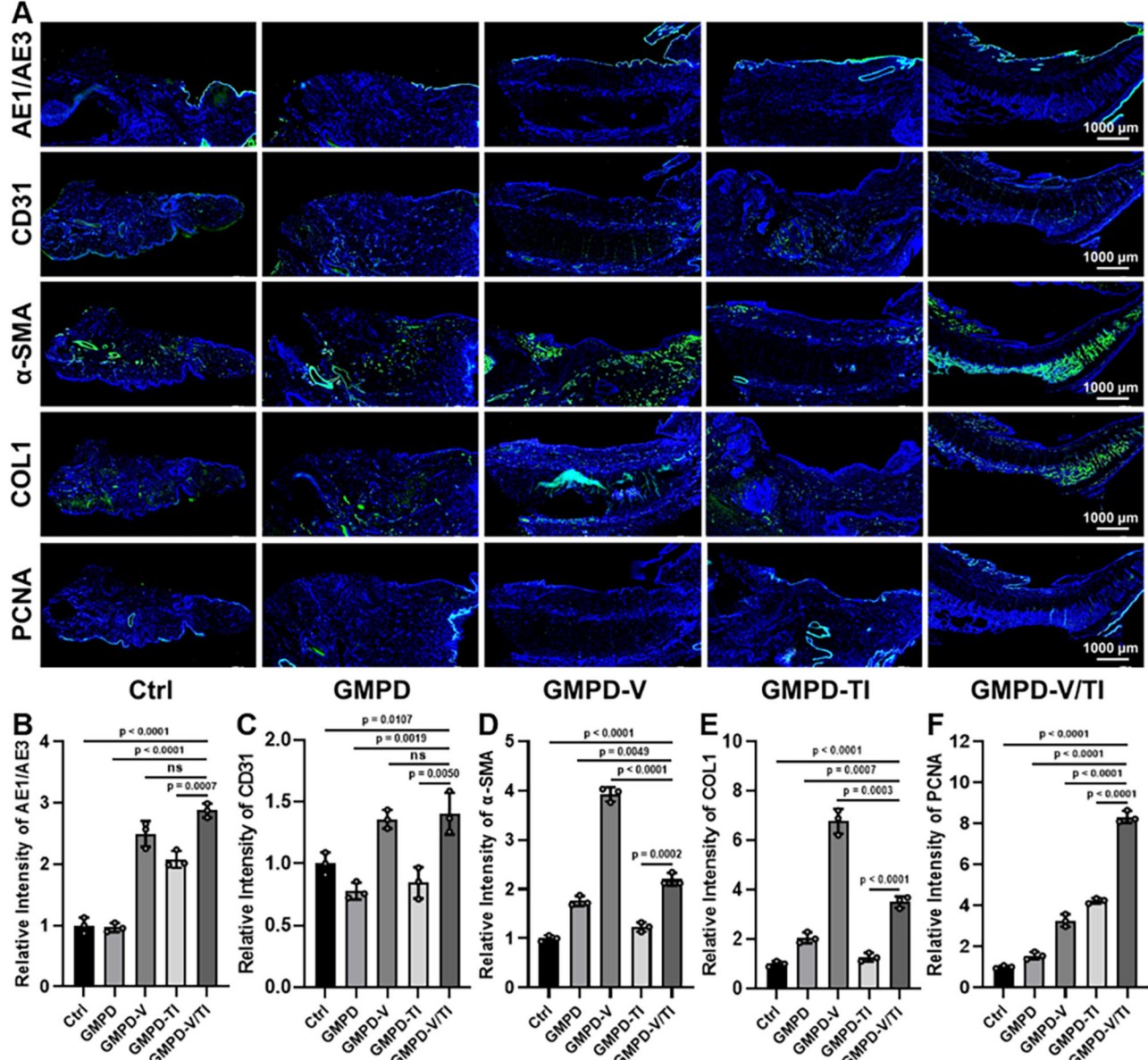

**Fig. 8 | Immunofluorescent examinations of scarless wound healing of the rabbit urethral canal. A** Immunofluorescence staining of the rabbit urethral canal for evaluating epithelialization (AE1/AE3), angiogenesis (CD31), fibrogenesis (α-SMA and COL1), and cell proliferation (PCNA) after different treatments (i.e., **GMPD, GMPD-V, GMPD-TI, GMPD-V/TI,** and **Ctrl** groups) for 8 weeks. **B**–**F** Quantitative expression levels of AE1/AE3 (**B**), CD31 (**C**), α-SMA (**D**), COL1 (**E**), and PCNA (**F**) after different treatments (i.e., **GMPD, GMPD-V, GMPD-TI, GMPD-V/TI,** and **Ctrl** groups) for 8 weeks. The hydrogel dressing compositions are the same as in Fig. 7. $n = 3$ biologically independent samples. Data are presented as mean ± SD. All error bars represent SD. $p$ values calculated using one-tailed unpaired $t$-test. ns = no significance. Source data are provided as a Source Data file.

repair, but clinical application is difficult due to ethical, cell colonization, and local survival problems[45,46]. Thus, current studies have focused on delivering biological factors with biocompatible scaffolds, especially hydrogel dressings, owing to the advantages of wet-healing conditions, the ease of carrying bioactive substances, and the availability to mimic the cell microenvironment. Predictably, there is high clinical value in developing ideal hydrogel dressings for scarless urethral reconstruction. To date, it still has limited breakthrough to efficiently promote wound healing against urethral stricture using the current hydrogel-based treatment. Therefore, a better urethral repair strategy needs to construct a time-dependent 4D modulation to fit well with different healing stages to balance ECM deposition and remodeling for scarless urethral reconstruction especially in the harsh UME.

The primary consideration for urethral repair is to overcome the harsh UME of dynamic wound surfaces suffering from acidic urine. In

this study, a hybrid crosslinking strategy combining dynamic boronic ester crosslinking and covalent photopolymerization was synergistically employed to prepare **GMPD** hydrogels. The hybrid design rationale is to obtain optimal physicochemical properties adaptable to harsh UME: (i) boronic ester crosslinking has good shear-shinning and self-healing features that are suitable for injecting onto the target site and completely covering the urethral defects, followed by secondary photopolymerization for post-stabilization; (ii) different from traditional elastic hydrogels, our hybrid designed hydrogels exhibit satisfactory mussel-mimetic viscoelasticity accessible to dynamic urethral environment; (iii) the inherent *cis*-diol-based adhesive ability derived from phenylboronic acid-modified **GM** together with further mechanical enhancement by covalent photopolymerization; (iv) the choice of fluorophenylboronic acid moiety is mainly related to physiological-pH crosslinking conditions and the acid-reinforced

mechanical strength that could effectively resist acidic urine environment. Compared with the commercially available hydrogels (e.g., Jelmyto and Backstop) approved for urinary applications that are easily washed away within hours after injection, our established hybrid crosslinking strategy possessed the advantages of facile operation, mussel-mimetic viscoelasticity, satisfactory adhesion, and acid-reinforced stability adapted to the harsh UME for long-lasting urethral wound protection.

Furthermore, the 4D design of hydrogel dressings plays a key role in achieving scarless urethral reconstruction. Fully mimicking the time-dependent physiological process requires an effective balance between ECM deposition and remodeling at different healing stages. The current results demonstrated that the **TOR** technical platform could simultaneously improve early-stage angiogenesis through **V-GM** microgels and prevent later-stage excessive fibrogenesis through **TI-PLGA** microcapsules. At the early stage, we found that seven genes were dominant in the wound healing process, including three upregulated DEGs (*VEGFA*, *HMOX1*, and *ITGA2*) and four downregulated DEGs (*HIF-1A*, *TPM1*, *CCN1*, and *COL1A1*), which are related to angiogenesis and urothelial regeneration. At the later stage, fibrogenic gene expression (*α-SMA*, *elastin*, *COL1*, *COL3*, and *fibronectin*) was significantly downregulated due to the negative regulation of TGFβ2/Smad3 signaling (*TGFB2* and *Smad3* downregulated) associated with the activation of MMP-related matrix degradation (*MMP1* upregulated). Taken together, the time-dependent physiological process of urethral reconstruction was successfully achieved through preferentially promoting vascularized urothelium regeneration by activating the VEGF signaling pathway and subsequently preventing excessive fibrogenesis by timing TGFβ2/Smad3 signaling pathway inhibition. The same effects of **TOR**-functionalized hydrogel dressing were further verified in a rabbit urethral injury model. Only the synergetic function of both VEGF and TGFβ inhibitor in the **GMPD-V/TI** group exhibited the optimal repair effect, showing satisfactory urothelium regeneration without hypertrophic scar formation.

In summary, the current study demonstrates a **TOR**-functionalized hydrogel dressing for scarless urethral reconstruction. The UME-adaptable **GMPD** hydrogels possess robust adhesion to dynamic wound surfaces even when suffering from acidic urine, while the **TOR** platform effectively balances ECM deposition and remodeling at different healing stages. As a result, the urethral injury of rabbits was sucessfully repaired through the time-dependent physiological process of preferentially activating VEGF-related vascularized urothelium regeneration and subsequently preventing TGFβ2/Smad3-related hypertrophic scar formation. Of course, there is still a long way to go for the future clinical translation based on our established 4D hydrogel dressing technique because some essential questions need to be investigated in a pig animal model, such as the comprehensive evaluation of material biosafety and foreign body inflammatory response, as well as the systematic exclusion of possible additional complications (e.g., the risk of urinary infection and urinary stone). Although this work represents a proof-of-concept study, we are fully convinced of the significance of time-dependent physiological urethral reconstruction, which paves the way to improve the clinical treatment of urethral injury in the future.

## Methods
### Materials and animals
Gelatin (from porcine skin), methacrylic anhydride, sodium hydroxide, 4-carboxy-3-fluoro-phenylboronic acid, D-(+)-gluconic acid δ-lactone, 1-ethyl-3-(3-dimethylaminopropyl) carbodiimide hydrochloride (EDC·HCl), N-hydroxy succinimide (NHS), triethylamine, dimethylsulfoxide (DMSO), poly(ethylene glycol) (PEG, MW: 400), polyvinyl alcohol (PVA, MW: 95, 000), and lithium phenyl-2,4,6-trimethylbenzoylphosphinate (LAP) photoinitiator were purchased from Sigma-Aldrich. Poly(lactic-co-glycolic acid) (PLGA, MW: 10 kDa, LA:GA = 75: 25) were

purchased from Daigang Biotechnology Co., Ltd. The TGFβ inhibitor SB431542 was purchased from Selleck. All the other chemicals were reagent grade. New Zealand white rabbits were purchased from Shanghai Jiao Tong University School of Agriculture. All protocols for animal experiments were approved by the Animal Care and Experimental Committee of Shanghai Jiao Tong University Affiliated Sixth People's Hospital (No: 2021-0154).

### Synthesis of GMP and GMD polymers
First, gelatin methacryloyl (**GM**) polymers were synthesized according to the previously established methods[47]. Briefly, 10 g of gelatin was dissolved in 500 mL of PBS, and then 10 mL of methacrylate anhydride was slowly added into the above solution for reacting 2 h. After the reaction, the insoluble substances were removed by centrifugation and the liquid supernatant was collected. The crude products were dialyzed against deionized water at 40 °C for 3 days followed by freezing and lyophilizing (92% yield). Then, 2 g of **GM** was dissolved in 50 mL of PBS (pH = 7.4) and stirred vigorously at 50 °C until complete dissolution. To synthesize **GMP** polymers, 92 mg (0.5 mmol) of 4-carboxy-3-fluoro-phenylboronic acid, 96 mg (0.5 mmol) of EDC·HCl, and 58 mg (0.5 mmol) of NHS were dissolved in anhydrous DMSO and sequentially added into the above **GM** solution. To synthesize **GMD** polymers, 140 mg (0.8 mmol) of D-(+)-gluconic acid δ-lactone and 0.1 mL of triethylamine were dissolved in anhydrous DMSO and sequentially added into the above **GM** solution. After the reaction, the liquid supernatant was collected and dialyzed against deionized water at 40 °C for 3 days followed by freezing and lyophilization (**GMP**: 87% yield; **GMD**: 73% yield). $^1$H NMR spectra were obtained to characterize the grafted functional groups and determine the corresponding substitution degree as previously described[19,48]. Briefly, the methacryloyl-substituted degree was determined by the decrease of proton peak at 2.9 ppm (amino groups of gelatin); the FPBA-substituted degree was determined by the integral ratio of the proton peaks at 7.3–7.5 ppm to the peaks at 5.2–5.7 ppm (methacryloyl groups); the diol-substituted degree was determined by the integral ratio of the proton peaks at 3.1 ppm to the peaks at 5.2–5.7 ppm (methacryloyl groups).

### Synthesis of V-GM microgels and TI-PLGA microcapsules
To produce **V-GM** microgels, 500 mg of **GM** and 20 mg of LAP were dissolved in 10 mL of PBS (pH = 7.4) containing 0.1 μg/mL VEGF (Dima Biotech) as the water phase. Then, 5 mL of span-80 and 40 mL of paraffin oil were mixed with each other as the oil phase. Both water and oil fluids were heated at 40 °C and injected into the microchannels by micropumps, in which the water phase formed single spherical droplets under the fluid shear of the oil phase. Subsequently, the droplets were crosslinked by photopolymerization under light irradiation (365-nm LED, 20 mW/cm²). The microgels were washed with hexane and deionized water for three times, followed by freezing and lyophilization. The **TI-PLGA** microcapsules were produced by the W/O/W emulsion method according to the previous literature[49]. Briefly, 1 mg of TGFβ inhibitor was added into a solution of 2.5 mg PEG400/50 μL deionized water to obtain the inner water phase. The above solution was added dropwise into a solution of 100 mg PLGA/1 mL chloroform with vigorous stirring for 10 min emulsification. The resultant water-in-oil (W/O) emulsion was then added dropwise into 15 mL of 1% w/v PVA aqueous solution as an emulsion stabilizer. The double emulsion (W/O/W) was obtained using a magnetic stirrer (3000 rpm) and allowed to stand for 5 h until solvent evaporation. The **TI-PLGA** microcapsules after centrifugation were washed three times, followed by freezing and lyophilization. The encapsulation efficiencies (EE) of **V-GM** microgels and **TI-PLGA** microcapsules were determined using the solvent extraction technique. Briefly, the concentration of extracted VEGF was analyzed by ELISA assay and the concentration of extracted TGFβ inhibitor was analyzed by ultraviolet-visible spectrophotometry. Then, EE value was calculated by the equation: EE (%) = actual drug content/theoretical

drug content. The drug release kinetics of both **V-GM** microgels and **TI-PLGA** microcapsules in **GMPD+*hv*** hydrogels were quantified using a standard curve (ELISA assay for VEGF release; ultraviolet-visible spectrophotometry for TGFβ inhibitor release) and normalized to total cumulative release at different time points.

## Hydrogel preparation

The hydrogel precursors of **GM, GMP, GMD,** and LAP (0.2% w/v) were mixed as a certain proportion in PBS solution (pH = 7.4). The hydrogel composition in this study was as follows: **GMP** hydrogels: 10% w/v FPBA-modified GelMA; **GM** hydrogels: 10% w/v GelMA; **GMPD** hydrogels: 10% w/v, **GMP: GMD** = 1:1; **GMPD-*hv*** hydrogels: **GMPD** hydrogels without light irradiation; **GMPD+*hv*** hydrogels: **GMPD** hydrogels with light irradiation; light: 365-nm LED, 20 mW/cm². The light irritation time was determined referred to the time sweep rheological results of reaching the complete crosslinking after ~30 s light irradiation.

## *XPS* and ATR-FTIR experiments

First, the **GMP** and **GMD** polymers, as well as the **GMPD-*hv*** and **GMPD+*hv*** hydrogels, were dried at 40 °C for 12 h. Then, the dry film samples were tested in an ultrahigh vacuum chamber by an ESCALAB 250Xi *XPS* system. The *XPS* spectra were analyzed by a XPSPEAK software to conduct peak separation. In addition, the dry film samples were also analyzed on a Nicolet 6700 FTIR spectrometer to obtain FTIR spectra.

## Rheological measurements

Dynamic rheology experiments were conducted on a HAAKE MARS III photorheometer at room temperature, using a 20-mm diameter parallel-plate geometry (P20 TiL) and OmniCure Series 2000 light source (365 nm, 20 mW/cm²). Strain sweeps were firstly performed to verify that the measuring parameters were in the linear region. Time sweep oscillatory tests were performed under the following measuring parameters: 10% strain; 1 Hz frequency; 0.5 mm gap; 120 s light irradiation. The gel point was considered when the storage modulus (G′) surpassed the loss modulus (G″). The final elastic modulus was determined as the storage modulus (G′) reaching complete gelation. Frequency sweep oscillatory tests were performed under the following measuring parameters: 10% strain; 0.5 mm gap; from 0.5 to 100 rad/s. Viscosity tests were performed under a gradually increasing shear rate (from 0 to 50 s⁻¹).

## SEM examination

To investigate the interfacial integration between hydrogels and muscle tissue, the hydrogel-muscle integrated samples were dehydrated by free drying and cut into a relatively flat interface for SEM examination. Then, the dehydrated samples were coated with gold-palladium in a Hitachi S-3400N ion sputter for morphological observation.

## Adhesion tests

For lap shear tests, fresh hot casing was attached to a glass slide with cyanoacrylate glue to prepare the testing samples. The **GMPD** hydrogel precursors were uniformly dispersed on the surface of a hot casing with or without light irradiation (365-nm LED, 20 mW/cm²). For incision sealing tests, porcine muscle was cut into 2 × 4 cm pieces, and followed by creating a 1-cm incision in the middle of the muscle tissue. Then, the **GMPD** hydrogel precursors were introduced into the muscle defects with or without light irradiation (365-nm LED, 20 mW/cm²). The adhesion experiments were performed on an Instron machine in a tensile mode at a 5 mm/min speed until the breakage of samples.

## Stability tests

The hydrogel samples after complete swelling were recorded as the initial weight $W_0$. Then, the hydrogel samples were immersed in PBS solution (pH = 7.4). At each time point, these samples were carefully collected and recorded as dry weight for testing hydrogel stability.

## In vitro biological evaluation

For cell scratch assay, both fibroblasts and endothelial cells were manually scratched with 200 µL pipette tips. Then, the wound area was observed by optical microscopy after 0, 6, 12, and 24 hours in the **V-GM, TI-PLGA**, and **Ctrl** groups. For cell proliferation experiments, cell viability was evaluated using a CCK-8 kit (Dojindo) according to the manufacturer's protocol. The optical density (OD) was measured using a microplate reader (Synergy H1, BioTek). For cell expression evaluation, the corresponding fibrogenic and angiogenic evaluations were examined by immunofluorescence staining. The fibrogenic expression levels of fibronectin, α-SMA, COL1, and COL3 in fibroblasts and the angiogenic expression levels of CD31 and VWF in endothelial cells were evaluated respectively. The statistical data of the relative fluorescent intensity were analyzed using ImageJ software.

## Sequencing read preprocessing

All low-quality bases and adapters of sequencing reads were trimmed using Trimmomatic (PMID: 35037208). After quality control, the reserved clean reads were mapped to the genome references of *Homo sapiens* (UCSC hg38) and *Oryctolagus-cuniculus* (Ensembl OryCun2.0) using STAR (PMID: 26334920). The gene counts and TPM (transcripts per million) values were calculated using RSEM (PMID: 21816040). The gene symbols corresponding to Ensembl gene IDs of *Oryctolagus-cuniculus* were retrieved from the UniProt database (PMID: 33237286). All DEGs were identified by R package DESeq2 (PMID: 25516281). An absolute value |log2FoldChange| of ≥1 and padj of ≤0.05 were considered statistically significant.

## Functional enrichment of DEGs

The enriched terms of Gene Ontology (GO) and Kyoto Encyclopedia of Genes and Genomes (KEGG) were summarized by using the R package clusterProfiler (PMID: 34557778). The top 15 enriched terms were plotted by using the R package ggplot2. In addition, all genes associated with wound healing (GO:0042060) were retrieved from the UniProt database (PMID: 33237286) according to their annotation.

## Gene ontology (GO) and KEGG enrichment analysis

Gene Ontology (GO; http://www.geneontology.org) is a systematic approach for gene and protein annotation in terms of biological process, molecular process, and cellular component. Kyoto Encyclopedia of Genes and Genomes (KEGG; http://www.genome.jp/kegg/) is an online database depositing biological pathways of genes and biochemicals. The enriched GO terms and KEGG pathways were annotated using the R package clusterProfiler.

## qRT-PCR tests

The total RNA of cell-hydrogel samples was firstly isolated using TRIzol reagent (Life Technologies). Then, the reverse transcription was conducted using a cDNA synthesis kit (Thermo Scientific) according to the manufacturer's instructions. The gene expression was quantitatively analyzed by SYBR Green using a 7500 Real-Time PCR system (Applied Biosystems, Life Technologies). The total primers and probes for detecting *VEGFA, HMOX1, ITGA2, HIF−1A, TPM1, CCN1, TGFBR2, Smad3, MMP1, α-SMA, elastin, COL1, COL3, fibronectin*, and *β*-actin were designed based on the published gene sequences (NCBI and PubMed). The expression level for each gene was normalized to *β*-actin.

## Western blot tests

The cell-hydrogel samples were harvested, treated by benzosulfonyl-fluoride, and then a protease and phosphatase inhibitor were added to dissolve the above samples in RIPA lysis buffer. The extracted cell lysates were cultured on ice (~3 h) and centrifuged to collect the supernatant. A 10% SDS-PAGE gel was used for electrophoresis with ~60-µg protein loaded in each lane. After the progress, the target proteins

were transferred to 0.45-μm polyvinylidene fluoride membranes (PVDF) and then blocked with 5% blocking buffer at 37 °C for 1 h. The PVDF membranes were treated with primary antibodies against VEGFA, ITGA2, HIF-1A, TGFBR2, Smad3, MMP1, α-SMA, elastin, COL1, COL3, fibronectin, and β-actin at 4 °C overnight. After that, the PVDF membranes were rinsed with TBST three times and incubated with the corresponding secondary antibody for 1 h. The membranes were scanned by an optical microscopy to present the expression of proteins.

### Urethroplasty and postoperative examinations in rabbit
To verify the feasibility of scarless urethral reconstruction, the hydrogel dressings were in situ formed in the urethral defects of rabbits for in vivo experiments. All animal experiments were performed in accordance with the guidelines for animal care. The animal protocol (SYXK 2017-0240) was approved by the Institutional Animal Care and Use Committee of the Shanghai Jiao Tong University Affiliated Sixth People's Hospital. Fifteen adult male New Zealand white rabbits with an average body weight of 2.5 kg were randomly divided into five groups for the creation of urethral defects. The rabbits were first subjected to general anaesthesia with intravenous injection of pentobarbital, and then the rabbits' skin and urethras were disinfected with 70% alcohol. The skin and ventral urethra were sectioned at ~3 cm proximal to the external urethral orifice, and the urethral lumen was exposed. A dorsal urethral defect with a mean length × width of 2.0 cm × 0.8 cm was created in the anterior urethra of rabbits. All the rabbits underwent removal of the urethra near the corpus cavernosum. Rabbits in group 1 ($n = 3$) were not repaired as control. Rabbits in group 2 ($n = 3$) were repaired with **GMPD** hydrogels. Rabbits in group 3 ($n = 3$) were repaired with **GMPD-V** hydrogels. Rabbits in group 4 ($n = 3$) were repaired with **GMPD-TI** hydrogels. Rabbits in group 5 ($n = 3$) were repaired with **GMPD-V/TI** hydrogels.

### Urethrography
To observe the urethral leakage and stricture in the five groups of rabbits, the contrast solution was firstly injected into the urethral lumen after 4- and 8-weeks post-surgery. Then, the rabbits underwent the urethral contrast-enhanced ultrasound measurement to check the degree of scar formation in the urethra. The rabbits were euthanized after retrograde urethrograms and the urethral tissue for the following histology staining were collected. The blockage ratio was semi-quantitatively estimated by the range of strictured urethra to the normal urethra according to the urethrography.

### Histology and immunofluorescence assessment
The urethral tissue was harvested after 8-weeks post-surgery for histology and immunofluorescence assessment. The specimens were firstly fixed in 4% paraformaldehyde for 30 min at room temperature. Then, they were dehydrated with different grades of alcohol and embedded in paraffin blocks. Histological sections were prepared and observed using an optical microscope. Hematoxylin and eosin staining (H&E) and Masson's trichrome staining tests were conducted to identify the epithelial layer and collagen distribution of the urethra. To further demonstrate the reconstruction of urethral function, the samples were stained for immunofluorescence for epithelial cytokeratin AE1/AE3 (Santa Cruz Biotechnology, Inc.), CD31 (Proteintech Group, Inc.), α-smooth muscle actin (Proteintech Group, Inc.), COL3 (Santa Cruz Biotechnology, Inc.), En1 (Santa Cruz Biotechnology, Inc.), CD206 (Proteintech Group, Inc.), and PCNA (Proteintech Group, Inc.). Nuclei were stained with DAPI (1:500, Life Technologies). Afterwards, these specimens were carefully imaged and observed by an optical microscope.

### Statistics and reproducibility
All data are presented as the means ± SDs. All error bars represent SD. Differences between the values were evaluated using one-tailed unpaired $t$-test with $P < 0.05$ considered statistically significant. Source data are provided as a Source Data file.

### Reporting summary
Further information on research design is available in the Nature Portfolio Reporting Summary linked to this article.

## Data availability
The authors declare that all data supporting of results in this study are available within the paper and its Supplementary Information, or from the corresponding authors upon request. Source data are provided with this paper.

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

## Acknowledgements

This research was financially supported by Jiangsu Key Technology Research Development Program (BE2017664), Shanghai Jiao Tong University Biomedical Engineering Cross Research Foundation (YG2022ZD022 and YG2017QN15), National Natural Science Foundation of China (82072217 and 81772135), Shanghai health committee (20184Y0053), Shanghai "Rising stars of medical talent" Youth development program, Shanghai Jiao Tong University K. C. Wong Medical Fellowship Fund, Shanghai sixth people's hospital foundational research program, National Key Research and Development Program of China (2022YFA1207500, 2018YFA0703100), and Shanghai Municipal Key Clinical Specialty (shslczdzk06601).

## Author contributions

Y.J.H. and K.Z. provided the idea. Y.J.H., Q.F., W.C., and K.Z. designed the experiments. Y.J.H., K.W., Y.Y.H., Y.W., W.F., Y.S., and K.Z. performed the experiments. Y.J.H., K.W., Y.Y.H., and K.Z. analyzed the data. Y.J.H. and K.Z. wrote the manuscript. Y.Z., G.Z., Q.F., and W.C. revised the manuscript. All authors commented on the manuscript and its revisions.

## Competing interests

The authors declare no competing interests.

## Additional information

[1]Department of Urology, Shanghai Sixth People's Hospital Affiliated to Shanghai Jiao Tong University School of Medicine, Shanghai Jiao Tong University,
Shanghai 200233, P. R. China. [2]Department of Plastic and Reconstructive Surgery, Shanghai Ninth People's Hospital Affiliated to Shanghai Jiao Tong
University School of Medicine, Shanghai Key Laboratory of Tissue Engineering, Shanghai 200011, P. R. China. [3]Clinical Research Center, Shanghai Chest
Hospital, Shanghai Jiao Tong University School of Medicine, Shanghai 200030, P. R. China. [4]Department of Orthopaedics, Shanghai Institute of Traumatology
and Orthopaedics, Shanghai Key Laboratory for Prevention and Treatment of Bone and Joint Diseases, Ruijin Hospital, Shanghai Jiao Tong University School of
Medicine, Shanghai 200025, P. R. China. [5]These authors contributed equally: Yujie Hua, Kai Wang, Yingying Huo. ✉e-mail: jamesqfu@126.com;
wgcui@sjtu.edu.cn; great_z0313@sjtu.edu.cn

