## [Peer Review File · Nature Communications]

Reviewers' Comments:

Reviewer #1:

Remarks to the Author:

In this manuscript, Hua et al. developed an adhesive hydrogel for urethral injury treatment via modulating the early-vascularized microenvironment and the later-antifibrogenic microenvironment in a time-dependent manner. The incorporation of VEGF-loaded gelatin methacryloyl (GM) microgels and TGF β inhibitor-loaded poly-lactic-co-glycolic acid (PLGA) microcapsules into the GMPD hydrogel facilitated wound healing by activating the VEGF signaling pathway and prevented hypertrophic scar formation by timing TGF β signaling pathway inhibition. The overall study is well organized. However, there are still some important concerns that should be addressed.

Major:

1. The delivery of VEGF for urethral injury treatments has been extensively explored (*Acta Biomaterialia*, 2020, 102: 247-258; *Biomaterials*, 2013, 34(34): 8617-8629). In addition, the scarless urethral reconstruction through TGF β signaling pathway was also confirmed (*Advanced Materials*, 2022, 34(14): 2109522). The combination of VEGF and TGF β inhibitor is believed to be a general approach for urethra repair. The novelty of this approach should be comprehensively and thoroughly discussed.
2. The temporally controlled drug delivery is essential for tissue regeneration. How did the authors determine the dosing interval of VEGF and TGF β inhibitor?
3. Even though the authors demonstrated the adhesive property of GMPD hydrogel in vitro, the in vivo adhesive performance should also be investigated since flow-induced shear force could wash away the hydrogel. Meanwhile, the retention behavior of GMPD hydrogel in the urethra should be evaluated.
4. What are the loading efficiencies of VEGF and TGF β inhibitor in the GM microgels and PLGA microcapsules, respectively?
5. The Authors could elaborate on the advantages of the proposed approach over hydrogels that are currently used or are in development. Limitations of the proposed approach should be mentioned.

Minor:

1. In Figure 3F and I, the sample numbers should be added.
2. The quantitative assay of Masson staining in Figure 7E should be conducted.
3. The scale bars in Figure 4E, 4H, and 7E are not clear.

Reviewer #2:

Remarks to the Author:

The authors described an elegant platform to enhance urethral reconstruction by using a polymeric hydrogel that has dual drug delivery functionality. The manuscript suits the needs of the fields of urology and biomaterials science and engineering. This work is highly innovative, the experimental methods sound and the results potentially beneficial for scarless urethral reconstruction. However, there are a few points that prohibit publication as is and require major revisions and explanations.

1. The authors claim that the harsh UME acidic urine is the main challenge in hydrogel-based urethral reconstructions. In this manuscript, a pH of 6.5 was considered "acidic urine" and pH-dependent experiments were conducted at pH 6.5, pH 7.4 and pH 8.5. These values are far from

being "acidic urine" since urine pH encompasses pH 4.5 to pH 8.0. In fact, acidic urine pH 5.0 and pH 5.5 are very common in healthy humans. pH-dependent experiments need to be conducted with significant values ranging from 4.5-8.

2. In line 201, the authors claim that "the in vitro degradation time from ~3 days (pH = 7.4) to ~14 days (pH = 5.0)" and refer the reader to Suppl. Fig. 4 which shows only degradation curves for pH 7.4 and pH 6.5. Thus, the experimental figure does not fit the conclusions shared in the manuscript. Additional experiments at pH 4.5-8 need to be conducted to evaluate the stability of the gels in real acidic urine.

3. The gel showed less than 20% degradation in 14 days in solution at pH 6.5 due to the acid-reinforced stability. The reviewer is concerned about the long-term stability of the gel in actual acidic urine (pH 5-6). The hydrogel is supposed to release its bioactive cargo within 7 days according to drug release experiments.

3.1. The authors should comment on how the gel will be cleared from the urinary tract after its action has been completed if the degradation is too slow. Most gels approved for intravesical applications (e.g., Jelmyto, Backstop, etc) are degraded within hours after injection. Solid (or semi-solid gels) and foreign bodies are excellent support for bacterial adhesion and biofilm formation and/or urinary stone nidus. While bacterial adhesion and crystal formation experiments may not be required at this time for the proof-of-concept, the authors should mention this as a potential limitation of the proposed platform.

4. Urethral injury promotes local tissue swelling that can cause acute urinary retention. The application of a hydrogel in a narrow channel such as the urethra could lead to urinary retention that could lead to additional complications that outweigh the benefit of having a scarless recovery, for example hydronephrosis.

4.1. The authors should include additional information about the in vivo experiments, such as urination of the animal and hydrogel remaining after the 4-week and 8-week period. If no hydrogel was observed in the 4-week and 8-week mark, when was the gel degraded?

4.2. Swelling experiments of the hydrogel in different pH environments is needed to understand the potential of the gel to increase its volume that could lead to urine obstruction.

5. How was the drug release monitored? Were the microcarriers included inside of the GM matrix or the release was monitored with the microcarriers in solution? Is the release dependent on pH?

5.1. How do the microcarriers themselves diffuse (leak) from the GM hydrogel matrix? The SEM image of Fig. 3C shows that the hydrogel has macropores of sizes significantly larger than the size of the microcarriers which could lead to premature leakage of microparticles, particularly the ones used for a delayed release.

5.2. Following up with the previous SEM images of the microcarriers inside of the gel matrix are interesting to obtain a better idea of the morphology of the designed platform.

6. In Figure 7C, could the authors provide additional information about blockage ratio and how do they estimate it? Why is it so high for the Ctrl, GMPD and GMPD-V?

7. The histological images need higher magnification. The authors make constant referrals to the urothelium and its regeneration but it is impossible to observe in detail the urothelial layer.

7.1. The reviewer is confused about the histology image showed for GMPD-TI. The lower part of the tissue has a perfect intact urothelium. However, the authors point at the top part of the image and claim that the urothelium has not been regenerated. Is it supposed to be urothelium layer there? From the low magnification image showed it seems that the authors are pointing to the muscle layer where no urothelium is supposed to be present. Please clarify. I apologize in advance if I am misreading the image.

8. Findings reported in Figure SI.10 have not been explained in detail. GMPD-TI shows a significantly lower inflammation than GMPD-V and GMPD-VTI. It seems like the presence of V-component in the platform induces inflammation. Please explain.

8.1 Is it acute or chronic inflammation? This is key to understand if the presence of a foreign body is causing it.

9. Irradiation with light induces photocrosslinking of the gel. To the best of my understanding, the time of irradiation has not been described in the paper. How was that time selected as the appropriate time of irradiation?

10. Authors have not included a limitations paragraph. Comments regarding clinical translation of this technology are needed.

Reviewer #3:

Remarks to the Author:

Zhang et al. submitted a manuscript entitled "Four-dimensional Hydrogel Dressing Adaptable to the Urethral Microenvironment for Scarless Urethral Reconstruction" to be considered as a research article in Nature Communications. This manuscript proposed a 4D hydrogel composed of reversible boronate linkage and irreversible covalent crosslinking by photo-irradiation. The synthetic hydrogel is designed to release VEGF and TGF β from V-GM and TI-PLGA respectively at different time points. The created time-dependent 4D microenvironment is claimed as a temporally on-demand regulatory (TOR) technical platform. The healing molecular mechanism has been performed to suggest potential genes for upregulation and downregulation in the wound healing process. An in vivo scarless urethral reconstruction has also been performed in rabbits to support the applicability of this hydrogel for promoting the regeneration of vascularized urothelium and the inhibition of hypertrophic scar formation by TOR functionalization. The principle and results are interesting, but there are some issues that need to be addressed as follows.

1. The proposed hydrogel's reversible linkage is boronate, which is sensitive to pH changes.

Authors claimed the dynamic crosslinking of boronic ester bonds exhibits acid-reinforced mechanism. However, the reviewer cannot find any information about pH value in the main text and figure legend (Figure 2H). The same problem can be found in the Figure 3B which doesn't reveal the pH value in the evaluation of tissue binding ability.

2. The research goal here is to provide a scarless urethral reconstruction for urethral injury. The proposed hydrogel should be used in the existing of urine. The pH range of urine could be in the range of 4.5-8 and is rich in urea. First, the only pH test in this manuscript I can find is Figure 2I and 2J which evaluated Wc and modulus in pH 6.5, 7.4, and 8.5. Authors should further check the material stability in lower pH conditions. Secondly, polycondensation is known to happen to boric acid and urea. May authors comment on this issue?

3. The chemical composition of GMP and GMDs well as GMPD-hv and GMPD+hv should be provided. The Mw and Mn of the polymers and the ratio of FPBA and cis-diol group are important to the readers.

4. Boronic acid could be sensitive under photo-irradiation with the generation of phenyl radical intermediate. Authors may have to check if side reactions can be found.

5. There are four XPS data including red, green, purple, and orange lines in each spectrum. Authors should provide details about what the color stands for in the figure legend.

Reviewer 1 comments and our corresponding answers:

Overall comment:

In this manuscript, Hua et al. developed an adhesive hydrogel for urethral injury treatment via modulating the early-vascularized microenvironment and the later-antifibrogenic microenvironment in a time-dependent manner. The incorporation of VEGF-loaded gelatin methacryloyl (GM) microgels and TGF β inhibitor-loaded poly-lactic-co-glycolic acid (PLGA) microcapsules into the GMPD hydrogel facilitated wound healing by activating the VEGF signaling pathway and prevented hypertrophic scar formation by timing TGF β signaling pathway inhibition. The overall study is well organized. However, there are still some important concerns that should be addressed.

Response:

Thanks for the reviewer's positive comments and professional suggestions on our work. According to these professional suggestions, we had tried our best to revise the manuscript and our responses point by point according to the reviewer's comments were listed as bellows.

Comments 1: The delivery of VEGF for urethral injury treatments has been extensively explored (Acta Biomaterialia, 2020, 102: 247-258; Biomaterials, 2013, 34(34): 8617-8629). In addition, the scarless urethral reconstruction through TGF β signaling pathway was also confirmed (Advanced Materials, 2022, 34(14): 2109522). The combination of VEGF and TGF β inhibitor is believed to be a general approach for urethra repair. The novelty of this approach should be comprehensively and thoroughly discussed.

Response 1: Many thanks for the reviewer's insightful comment. As the reviewer concerned, the previous researches have verified that the promotion of angiogenesis by VEGF delivery is essential for urethral regeneration, while the inhibition of TGF β /Smad signaling pathway facilitates the wound healing. In fact, the scarless urethral reconstruction needs to simultaneously mimic time-dependent physiological

processes, which requires balancing extracellular matrix deposition and remodeling at different healing stages. Thus, it is necessary to effectively promote early-stage vascularized urothelium regeneration and simultaneously prevent later-stage excessive fibrogenesis. To better clarify the novelty of combining VEGF and TGF β inhibitor, we have added the corresponding expression of “Predictably, there is high clinical value in developing ideal hydrogel dressings for scarless urethral reconstruction. To date, it still has limited breakthrough to efficiently promote wound healing against urethral stricture using the current hydrogel-based treatment. Therefore, a better urethral repair strategy needs to construct a time-dependent 4D modulation to fit well with different healing stages to balance ECM deposition and remodeling for scarless urethral reconstruction especially in the harsh UME.” and “Furthermore, the 4D design of hydrogel dressings plays a key role in achieving scarless urethral reconstruction. Fully mimicking the time-dependent physiological process requires an effective balance between ECM deposition and remodeling at different healing stages. The current results demonstrated that the TOR technical platform could simultaneously improve early-stage angiogenesis through V-GM microgels and prevent later-stage excessive fibrogenesis through TI-PLGA microcapsules.” in the discussion section.

Comments 2: The temporally controlled drug delivery is essential for tissue regeneration. How did the authors determine the dosing interval of VEGF and TGF β inhibitor?

Response 2: Many thanks for the reviewer’s professional suggestions. In fact, we determined the dosing interval of VEGF and TGF β inhibitor dependent on the previous research results on urethral reconstruction [References: Matthias, D. H. et al. Analysis of Primary Urethral Wound Healing in the Rat. *Urology* 84, 246.e1-e7 (2014); Shailesh, S. et al. Evaluation of healing at urethral anastomotic site by pericatheter retrograde urethrogram in patients with urethral stricture. *Urol. Ann.* 6, 325-327 (2014)]. The physiological urethral healing process usually needs to meet the requirements of early-stage urethral regeneration (3-5 days) and later-stage

anti-fibrogenesis function (7-14 days). The current results demonstrated that V-GM microgels exhibited initial burst release within ~3 days, while the core-shell structure of TI-PLGA microcapsules could effectively delay the release of the TGF β inhibitor after 3 days and sustain lasting release until 10 days, confirming the successful construction of the TOR platform by means of early-released V-GM microgels and later-released TI-PLGA microcapsules. To support our rationale of determining the dosing interval, we added the corresponding references as “As shown in Fig. 4D, V-GM microgels exhibited initial burst release within ~3 days, meanwhile the core-shell structure of TI-PLGA microcapsules could effectively delay the release of the TGF β inhibitor after 3 days and sustain lasting release until 10 days, which basically conformed to the time-dependent process of urethral reconstruction as previously reported^{41, 42}. Together, the TOR platform was successfully fabricated by means of early-released V-GM microgels and later-released TI-PLGA microcapsules.” in the results section.

Figure 1. (A) Schematic depiction of the process of dermal healing and (B) urethral healing in the rat. Urethral healing recapitulates the phases of dermal healing, however, with extension of each phase.

Comments 3: Even though the authors demonstrated the adhesive property of GMPD hydrogel in vitro, the in vivo adhesive performance should also be investigated since flow-induced shear force could wash away the hydrogel. Meanwhile, the retention behavior of GMPD hydrogel in the urethra should be evaluated.

Response 3: Many thanks for the reviewer’s professional suggestions. According to the reviewer’s suggestions, we have added the corresponding characterization of *in vivo* adhesive performance to investigate the effect of flow-induced shear force. The

corresponding expression of “More importantly, the GMPD+*hν* hydrogels could also withstand the flow-induced shear force *in vivo*, which is essential for adapting to the wet and dynamic urethral environment (Supplementary Fig. 4).” was added in the results section.

Supplementary Fig. 4. Representative photographs showing the resistance to water washing at approximately 10 kPa water pressure in the urethral defect model. Red arrows represent the adhesive hydrogels stained by green fluorescence.

In addition, the retention behavior of GMPD hydrogel in the urethra was also evaluated and the corresponding data were added in Supplementary Fig. 6. The corresponding expression of “The *in vivo* degradation experiments further showed the remaining hydrogels fully covered the urethral defects after 14 days, which could basically meet the requirement of wound healing process (Supplementary Fig. 6).” was added in the results section.

Supplementary Fig. 6. A, B) Representative photographs (A) and histological examinations of H&E and Masson's trichrome staining (B) of the remaining GMPD hydrogels in the urethral defect model after 3-, 7-, 14-days surgery. Blue circles and red arrows represent the remaining

hydrogels.

Comments 4: What are the loading efficiencies of VEGF and TGF β inhibitor in the GM microgels and PLGA microcapsules, respectively?

Response 4: Many thanks for the reviewer's professional suggestions. According to the reviewer's suggestions, we have added the characterization of the encapsulation efficiencies of VEGF (~58.7%) in the GM microgels and TGF β inhibitor (~83.6%) in the PLGA microcapsules. The corresponding expression of "As shown in Fig. 4A, a novel TOR technical platform was developed to address the above challenge by preferentially releasing VEGF from V-GM microgels (~58.7% encapsulation efficiency) and subsequently releasing a TGF β inhibitor (SB431542) from TI-PLGA microcapsules (~83.6% encapsulation efficiency)." was revised in the results section. In addition, the corresponding method description of "The encapsulation efficiencies (EE) of V-GM microgels and TI-PLGA microcapsules were determined using the solvent extraction technique. Briefly, the concentration of extracted VEGF was analyzed by ELISA assay and the concentration of extracted TGF β inhibitor was analyzed by ultraviolet-visible spectrophotometry. Then, EE value was calculated by the equation: $EE(\%) = \text{actual drug content} / \text{theoretical drug content}$." was added in the methods section.

Comments 5: The Authors could elaborate on the advantages of the proposed approach over hydrogels that are currently used or are in development. Limitations of the proposed approach should be mentioned.

Response 5: Many thanks for the reviewer's insightful comments.

According to the reviewer's suggestions, we added the two advantages of our proposed approach over other hydrogels as "Compared with the commercially available hydrogels (e.g., Jelmyto and Backstop) approved for urinary applications that are easily washed away within hours after injection, our established hybrid crosslinking strategy possessed the unique advantages of facile operation, mussel-mimetic viscoelasticity, satisfactory adhesion, and acid-reinforced stability

adapted to the harsh UME for long-lasting urethral wound protection.” and “Furthermore, the 4D design of hydrogel dressings plays a key role in achieving scarless urethral reconstruction. Fully mimicking the time-dependent physiological process requires an effective balance between ECM deposition and remodeling at different healing stages. The current results demonstrated that the TOR technical platform could simultaneously improve early-stage angiogenesis through V-GM microgels and prevent later-stage excessive fibrogenesis through TI-PLGA microcapsules.” in the discussion section.

According to the reviewer’s suggestions, we also added the potential limitations of our established technique for the future clinical translation as the supplemented expression of “Of course, there is still a long way to go for the future clinical translation based on our established 4D hydrogel dressing technique because some essential questions need to be investigated in a pig animal model, such as the comprehensive evaluation of material biosafety and foreign body inflammatory response, as well as the systematic exclusion of possible additional complications (*e.g.*, the risk of urinary infection and urinary stone). Although this work represents a proof-of-concept study, we are fully convinced of the significance of time-dependent physiological urethral reconstruction, which paves the way to improve the clinical treatment of urethral injury in the future.” in the discussion section.

Comments 6: In Figure 3F and I, the sample numbers should be added. The quantitative assay of Masson staining in Figure 7E should be conducted. The scale bars in Figure 4E, 4H, and 7E are not clear.

Response 6: Many thanks for your careful reviewing. According to the reviewer’s suggestions, we have added the sample numbers ($n = 4$) in Figure 3F and I. The semi-quantitative data of Masson’s trichrome staining in Figure 7E were added in Supplementary Fig. 13. The scale bars in Figure 4E, 4H, and 7E were further magnified to be clearer for readers.

Supplementary Fig. 13. The semi-quantitative data of Masson's trichrome staining of the scar thickness in the GMPD, GMPD-V, GMPD-TI, GMPD-V/TI, and control (Ctrl) groups after 8-weeks surgery.

Fig. 4 *In vitro* biological evaluation of the TOR platform. **A** Schematic of the design of the TOR platform by means of early-released V-GM microgels and later-released TI-PLGA microcapsules. **B** Representative microscope (i) and SEM (ii) images of V-GM microgels. **C** Representative SEM images of the intact (i) and crushed (ii) TI-PLGA microcapsules. Blue arrows represent the shell of PLGA microcapsules. Red arrows represent the cargo of TGFβ inhibitor powders. **D** The daily drug release curves of V-GM microgels and TI-PLGA microcapsules. **E** Observation of the fibroblast scratch assay after 0, 6, 12, and 24 hours in the V-GM, TI-PLGA, and control (Ctrl) groups. **F** Statistical analyses of the corresponding wound healing rates. **G** Cell proliferation rates of fibroblasts tested by CCK-8 assay after 1-, 4-, and 7-days culture in the V-GM, TI-PLGA, and Ctrl groups. **H** Immunofluorescence staining of fibronectin (red), α-SMA (green), and cell nuclei (DAPI, blue) for fibroblast marker expression after 4- and 7-days culture in the V-GM and TI-PLGA groups. **I-L** Comparative fibrogenic expression levels (fibronectin, α-SMA, COL1, and COL3) after 4- and 7-days culture in the V-GM, TI-PLGA, and Ctrl groups. TOR: temporally on-demand regulatory; TI-PLGA: TGFβ inhibitor-loaded PLGA microcapsules; V-GM: VEGF-loaded GelMA microgels. (n = 4, *p < 0.05, **p < 0.01, ***p < 0.001, ****p < 0.0001, #p < 0.001 compared with the corresponding group at day 4, ns = no significance)

Fig. 7 *In vivo* scarless urethral reconstruction using GMPD hydrogel dressings in rabbits. **A** Photographs of the surgical operation of *in situ* urethral defect repair using GMPD hydrogel dressings. **B, C** Urethrography images (**B**) and blockage ratios (**C**) of the rabbit urethral canal in the GMPD, GMPD-V, GMPD-TI, GMPD-V/TI, and control (Ctrl) groups after 4- and 8-weeks surgery. Red arrows represent urethral stricture. Blue arrows represent urethral patency. **D, E** Gross view (**D**) and histological examinations of H&E and Masson's trichrome staining (**E**) of the rabbit urethral canal in the GMPD, GMPD-V, GMPD-TI, GMPD-V/TI, and control (Ctrl) groups after 8-weeks surgery. GMPD hydrogels: 10% w/v GMPD; GMPD-V hydrogels: 10% w/v GMPD with 1% w/v V-GM; GMPD-TI hydrogels: 10% w/v GMPD with 1% w/v TI-PLGA; GMPD-V/TI hydrogels: 10% w/v GMPD with 1% w/v V-GM and 1% w/v TI-PLGA; light: 365-nm LED, 20 mW/cm², 30 s irradiation time. (n = 4, ****p < 0.0001)

Reviewer 2 comments and our corresponding answers:

Overall comment:

The authors described an elegant platform to enhance urethral reconstruction by using a polymeric hydrogel that has dual drug delivery functionality. The manuscript suits the needs of the fields of urology and biomaterials science and engineering. This work is highly innovative, the experimental methods sound and the results potentially beneficial for scarless urethral reconstruction. However, there are a few points that prohibit publication as is and require major revisions and explanations.

Response:

Thanks for the reviewer's great interests, positive comments, and professional suggestions on our work. According to these professional suggestions, we had tried our best to revise the manuscript and our responses point by point according to the reviewer's comments were listed as bellows.

Comments 1: The authors claim that the harsh UME acidic urine is the main challenge in hydrogel-based urethral reconstructions. In this manuscript, a pH of 6.5 was considered "acidic urine" and pH-dependent experiments were conducted at pH 6.5, pH 7.4 and pH 8.5. These values are far from being "acidic urine" since urine pH encompasses pH 4.5 to pH 8.0. In fact, acidic urine pH 5.0 and pH 5.5 are very common in healthy humans. pH-dependent experiments need to be conducted with significant values ranging from 4.5-8.

Response 1: Many thanks for the reviewer's insightful comment. According to the reviewer's suggestions, we have added the corresponding assessment of rheological properties and mechanical stability at pH 4.5. As shown in Fig. 2I, J and S5, the experimental results also showed the acid-reinforced mechanical strength and stability in the acidic urine environment. The corresponding expression of "Importantly, the dynamic crosslinking of boronic ester bonds exhibited acid-reinforced mechanical strength with ω_c increasing as pH decreased (ranging from 4.5-8.5), which is very suitable for harsh UME of low pH values (Fig. 2H, I). Moreover, time sweep

rheological measurements demonstrated that secondary photopolymerization could effectively improve the elasticity of GMPD hydrogels without any influence of pH values (Fig. 2J).” and “As shown in Supplementary Fig. 5, GMPD-*hv* hydrogels in acid solution could effectively extend the *in vitro* degradation time from ~3 days (*pH* = 7.4) to ~14 days (*pH* = 4.5), which is attributed to the acid-reinforced structural stability of boronic ester bonds (correlated with Fig. 2I). Additionally, GMPD+*hv* hydrogels exhibited long-lasting stability over 14 days in any harsh UME (pH values from 4.5 to 7.4) due to the pH-independent covalent stabilization of photopolymerization.” were revised in the results section.

Fig. 2I The crossover frequencies (ω_c) obtained from frequency sweep rheology for GMPD-*hv* hydrogels at each pH value ranging from 4.5-8.5. J Final shear moduli of GMPD+*hv* hydrogels at each pH value ranging from 4.5-8.5.

Supplementary Fig. 5. The *in vitro* degradation curves of GMPD-*hv* and GMPD+*hv* hydrogels in neutral (pH = 7.4) or acidic solutions (pH = 6.5 and 4.5).

Comments 2: In line 201, the authors claim that “the *in vitro* degradation time from ~3 days (pH = 7.4) to ~14 days (pH = 5.0)” and refer the reader to Suppl. Fig. 4 which shows only degradation curves for pH 7.4 and pH 6.5. Thus, the experimental figure does not fit the conclusions shared in the manuscript. Additional experiments at pH 4.5-8 need to be conducted to evaluate the stability of the gels in real acidic urine.

Response 2: Many thanks for your careful reviewing. Sorry for our mistake to miswrite the expression of “pH 6.5” as “pH 5.0” referred to Suppl. Fig. 4. According to the reviewer’s suggestions, we have added the corresponding assessment of mechanical stability at pH 4.5 (Supplementary Fig. 5). Also, we have revised the expression of “the *in vitro* degradation time from ~3 days (pH = 7.4) to ~14 days (pH = 4.5)” in the results section.

Comments 3: The gel showed less than 20% degradation in 14 days in solution at pH 6.5 due to the acid reinforced stability. The reviewer is concerned about the long-term stability of the gel in actual acidic urine (pH 5-6). The hydrogel is supposed to release its bioactive cargo within 7 days according to drug release experiments.

The authors should comment on how the gel will be cleared from the urinary tract after its action has been completed if the degradation is too slow. Most gels approved for intravesical applications (e.g., Jelmyto, Backstop, etc) are degraded within hours after injection. Solid (or semi-solid gels) and foreign bodies are excellent support for bacterial adhesion and biofilm formation and/or urinary stone nidus. While bacterial adhesion and crystal formation experiments may not be required at this time for the proof-of-concept, the authors should mention this as a potential limitation of the proposed platform.

Response 3: Many thanks for the reviewer’s insightful comments.

Regarding the mechanical stability: As the reviewer concerned, the long-term stability of the gel in the actual acidic urine (pH = 4.5 and 6.5) was further added in Supplementary Fig. 5. Actually, the gel presented a gradual degradation tendency within 14 days, in which the bioactive cargos could steadily release from the GMPD+*hν* hydrogels.

In addition, we also supplemented the *in vivo* degradation experiments in Supplementary Fig. 6. The corresponding expression of “The *in vivo* degradation experiments further showed the remaining hydrogels fully covered the urethral defects after 14 days, which could basically meet the requirement of wound healing process (Supplementary Fig. 6).” was added in the results section.

Supplementary Fig. 6. A, B) Representative photographs (A) and histological examinations of H&E and Masson's trichrome staining (B) of the remaining GMPD hydrogels in the urethral defects after 3-, 7-, 14-days surgery. Blue circles and red arrows represent the remaining hydrogels.

Regarding the issue of urethral infection: The urethral infection is a concern during the process of repair of urethra. In clinic, without the present hydrogel, the urinary catheter has to be kept in urethra for at least 3 weeks, which will often cause urinary infection and biofilm on catheter. Here, we used the hydrogel, which could accelerate the wound healing, the catheter could be removed earlier. Thus, the urinary infection and biofilm will be solved. On the other hand, with the early flowing of urine in urethra, the biotic and stone could be rinsed and cleaned. According to the reviewer's suggestions, we added this concern as “Of course, there is still a long way to go for the future clinical translation of our established 4D hydrogel dressing technique because some essential questions need to be investigated in a pig animal model, such

as the comprehensive evaluation of material biosafety and foreign body inflammatory response, as well as the systematic exclusion of possible additional complications (e.g., the risk of urinary infection and urinary stone).” in the discussion section.

Comments 4: Urethral injury promotes local tissue swelling that can cause acute urinary retention. The application of a hydrogel in a narrow channel such as the urethra could lead to urinary retention that could lead to additional complications that outweigh the benefit of having a scarless recovery, for example hydronephrosis.

The authors should include additional information about the *in vivo* experiments, such as urination of the animal and hydrogel remaining after the 4-week and 8-week period. If no hydrogel was observed in the 4-week and 8-week mark, when was the gel degraded?

Swelling experiments of the hydrogel in different pH environments is needed to understand the potential of the gel to increase its volume that could lead to urine obstruction.

Response 4: Many thanks for the reviewer’s insightful comments.

Regarding the risk of urinary retention: Actually, the GMPD+*hν* hydrogels didn’t show obvious swelling ratio in acidic pH value (Supplementary Fig. 7), meanwhile the hydrogel dressing presented an extremely thin layer (approximately 200 μm) to totally cover the urethral wound, which hardly cause the risk of urinary retention. In the animal models, a urinary catheter was kept in the urethra and stayed in the lumen of urethra for at least 2 weeks, which would decrease the possibility of urinary retention and its additional complications. Furthermore, the urethrography clearly presented a more fluent lumen in the GMPD-V/TI treated urethra than that in the non-treated group.

Regarding the *in vivo* degradation: According to the reviewer’s suggestions, we have supplemented the *in vivo* degradation experiments of hydrogels in urethral environment. The corresponding expression of “The *in vivo* degradation experiments further showed the remaining hydrogels fully covered the urethral defects after 14 days, which could basically meet the requirement of wound healing process

(Supplementary Fig. 6).” was added in the results section.

Regarding the swelling ratio: According to the reviewer’s suggestions, we have supplemented the swelling experiments of the hydrogels in different pH values. The corresponding expression of “In addition, GMPD+*hν* hydrogels didn’t show obvious swelling ratio in acidic pH value due to the secondary covalent photopolymerization (Supplementary Fig. 7).” was added in the results section.

Supplementary Fig. 7. The *in vitro* swelling ratio of GMPD+*hν* hydrogels in different pH values ranging from 4.5-8.5.

Comments 5: How was the drug release monitored? Were the microcarriers included inside of the GM matrix or the release was monitored with the microcarriers in solution? Is the release dependent on pH?

How do the microcarriers themselves diffuse (leak) from the GM hydrogel matrix? The SEM image of Fig. 3C shows that the hydrogel has macropores of sizes significantly larger than the size of the microcarriers which could lead to premature leakage of microparticles, particularly the ones used for a delayed release.

Following up with the previous SEM images of the microcarriers inside of the gel matrix are interesting to obtain a better idea of the morphology of the designed platform.

Response 5: Many thanks for the reviewer’s professional suggestions.

Regarding the test methods of drug release kinetics: According to the reviewer’s suggestions, we have added the test methods of drug release kinetics as “Briefly, the

concentration of extracted VEGF was analyzed by ELISA assay and the concentration of extracted TGF β inhibitor was analyzed by ultraviolet-visible spectrophotometry. The drug release kinetics of both V-GM microgels and TI-PLGA microcapsules in GMPD+*h* ν hydrogels were tested according to previously established methods⁴⁰.” in the methods section. In fact, the drug release kinetics were monitored with the microcarriers in the GMPD+*h* ν hydrogels. As a result, the drug release kinetics were pH-independent because of mechanically stabilizing the hydrogels after the secondary photopolymerization. To avoid any misleading, we also added the corresponding details of “Fig. 4D The daily drug release curves of V-GM microgels and TI-PLGA microcapsules in GMPD+*h* ν hydrogels.” in the figure legend.

Regarding the release principle and SEM images: In fact, as the reviewer’s concern, the microcarriers themselves can diffuse from the hydrogels in the lyophilized state, but they hardly leak from the hydrogels in the gel state until the degradation of hydrogels. It also needs to be noted that the larger pore morphology in Fig. 3C-i is the separated interface between the GM gel and mussel, while the pore morphology of GMP gel in Fig. 3C-ii present the relatively uniform porous structure. As the reviewer’s concern, it is a better idea to employ the different morphologies of the lyophilized hydrogels to design the TOR platform, which we will try to conduct this idea in the future research. Thanks again for the reviewer’s valuable suggestions.

Comments 6: In Figure 7C, could the authors provide additional information about blockage ratio and how do they estimate it? Why is it so high for the Ctrl, GMPD and GMPD-V?

Response 6: Many thanks for your careful reviewing. Sorry for the unclear description about blockage ratio. In fact, the blockage ratio was semi-quantitatively estimated by the diameter of strictured urethra to the normal urethra referred to the urethrography. According to the reviewer’s suggestions, we have added the additional information in the figure legend as “B, C Urethrography images (B) and the corresponding blockage ratios (C) of the rabbit urethral canal in the GMPD, GMPD-V, GMPD-TI, GMPD-V/TI, and control (Ctrl) groups after 4- and 8-weeks surgery.” and

“The blockage ratio was semi-quantitatively estimated by the diameter of strictured urethra to the normal urethra according to the urethrography.” in the methods section. In addition, the results of higher blockage ratio for the Ctrl, GMPD, and GMPD-V groups were attributed to the urethral stricture caused by the hypertrophic scar formation and thus occupied the urethral lumen.

Comments 7: The histological images need higher magnification. The authors make constant referrals to the urothelium and its regeneration but it is impossible to observe in detail the urothelial layer.

The reviewer is confused about the histology image showed for GMPD-TI. The lower part of the tissue has a perfect intact urothelium. However, the authors point at the top part of the image and claim that the urothelium has not been regenerated. Is it supposed to be urothelium layer there? From the low magnification image showed it seems that the authors are pointing to the muscle layer where no urothelium is supposed to be present. Please clarify. I apologize in advance if I am misreading the image.

Response 7: Many thanks for your careful reviewing. According to the reviewer’s suggestions, we have added the magnified histological images of epithelial cytokeratin AE1/AE3 staining in the supporting information to clearly show the urothelium regeneration. The corresponding expression of “The expression level of epithelial cytokeratin AE1/AE3, an important membrane surface protein marker, was significantly higher in the GMPD-V/TI groups than in the other groups. Specifically, the regenerated urothelium layer in the GMPD-V and GMPD-TI groups was thinner than that in the GMPD-V/TI group, whereas the expression of cytokeratin in both the Ctrl and GMPD groups was hardly found.” was revised in the main text.

Supplementary Fig. 14. The magnified immunofluorescence staining of the rabbit urethral canal for evaluating epithelialization (AE1/AE3) after different treatments (i.e., GMPD, GMPD-V, GMPD-TI, GMPD-V/TI, and Ctrl groups) for 8 weeks.

Sorry for the unclear histology images showed for GMPD-TI group. To avoid any misleading, we have provided the magnified histological images of epithelial cytokeatin AE1/AE3 staining in the revised manuscript. For the histological images, the upper and lower part of the tissues all showed a layer of cells at the place of urothelial layer. However, some cells in the upper part of the tissues are not mature due to the lack of AE1/AE3 expression. One reason might be that the cells have not differentiated to the mature urothelial cells. Thus, although the morphology seems like urothelial layer, but it lacks or low-expresses the marker and function of urothelial cells. For better observation, we magnified the histology images of the GMPD-TI group in the following figure.

GMPD-TI

Supplementary Fig. The magnified immunofluorescence staining of the rabbit urethral canal for evaluating epithelialization (AE1/AE3) in GMPD-TI group for 8 weeks.

Comments 8: Findings reported in Figure SI.10 have not been explained in detail. GMPD-TI shows a significantly lower inflammation than GMPD-V and GMPD-VTI. It seems like the presence of V-component in the platform induces inflammation. Please explain.

Is it acute or chronic inflammation? This is key to understand if the presence of a foreign body is causing it.

Response 8: Many thanks for the reviewer's insightful comments.

Regarding the CD206 staining and inflammation reaction: Actually, the CD206 immunofluorescence staining is a marker of M2 macrophages (the more CD206

positive staining, the less inflammation). It needs to be noted that M1 macrophages can secrete pro-inflammatory factors that have the role of phagocytic pathogens and apoptotic cells in the early stage of inflammation, while M2 macrophages can inhibit inflammatory response that further facilitate the tissue repair and reconstruction in the later stage of inflammation. The current results demonstrated that the CD206 staining was highly expressed in the groups of GMPD-V and GMPD-V/TI, which was conducive to urethral defect repair and reconstruction. Therefore, the presence of V-component in the current platform is beneficial to reduce the inflammation response. Of course, the underlying mechanism of V-component function in this urethral defect model needs to be further study in the future research. To avoid any misleading, we have revised the corresponding expression of “Different degrees of inflammatory responses at the injured sites appeared due to the application of the extra hydrogel dressings (Supplementary Fig. 15). Noticeably, the current results demonstrated that the number of CD206-positive cells (M2 macrophages labelled by CD206) was highly expressed in both GMPD-V and GMPD-V/TI groups than those of other groups, which were conducive to urethral defect repair because M2 macrophages could effective inhibit the inflammatory response and promote tissue regeneration.” in the results section.

Comments 9: Irradiation with light induces photocrosslinking of the gel. To the best of my understanding, the time of irradiation has not been described in the paper. How was that time selected as the appropriate time of irradiation?

Response 9: Many thanks for the reviewer’s professional suggestions. Actually, we chose the irradiation time of 30 s for photocrosslinking referred to the time sweep rheological results (reach the complete crosslinking after approximately 30 s light irradiation) as seen in Fig. 2G. According to the reviewer’s suggestions, we have added the irradiation time in the figure legend (Fig. 2 and Fig. 7) as “light: 365-nm LED, 20 mW/cm², 30 s irradiation time” and the corresponding selection reason of “The light irritation time was determined referred to the time sweep rheological results of reaching the complete crosslinking after approximately 30 s light irradiation”

was added in the methods section.

Comments 10: Authors have not included a limitations paragraph. Comments regarding clinical translation of this technology are needed.

Response 10: Many thanks for the reviewer's insightful comment that is beneficial to the improvement of our work regarding clinical translation in the future. According to the reviewer's suggestions, we have added the potential limitations of our established technique for the future clinical translation as the supplemented expression of "Of course, there is still a long way to go for the future clinical translation of our established 4D hydrogel dressing technique because some essential questions need to be investigated in a pig animal model, such as the comprehensive evaluation of material biosafety and foreign body inflammatory response, as well as the systematic exclusion of possible additional complications (*e.g.*, the risk of urinary infection and urinary stone). Although this work represents a proof-of-concept study, we are fully convinced of the significance of time-dependent physiological urethral reconstruction, which paves the way to improve the clinical treatment of urethral injury in the future." in the discussion section.

Reviewer 3 comments and our corresponding answers:

Overall comment:

Zhang et al. submitted a manuscript entitled “Four-dimensional Hydrogel Dressing Adaptable to the Urethral Microenvironment for Scarless Urethral Reconstruction” to be considered as a research article in Nature Communications. This manuscript proposed a 4D hydrogel composed of reversible boronate linkage and irreversible covalent crosslinking by photo-irradiation. The synthetic hydrogel is designed to release VEGF and TGF β from V-GM and TI-PLGA respectively at different time points. The created time-dependent 4D microenvironment is claimed as a temporally on-demand regulatory (TOR) technical platform. The healing molecular mechanism has been performed to suggest potential genes for upregulation and downregulation in the wound healing process. An in vivo scarless urethral reconstruction has also been performed in rabbits to support the applicability of this hydrogel for promoting the regeneration of vascularized urothelium and the inhibition of hypertrophic scar formation by TOR functionalization. The principle and results are interesting, but there are some issues that need to be addressed as follows.

Response:

Thanks for the reviewer’s great interests, positive comments, and professional suggestions on our work. According to these professional suggestions, we had tried our best to revise the manuscript and our responses point by point according to the reviewer’s comments were listed as bellows.

Comments 1: The proposed hydrogel's reversible linkage is boronate, which is sensitive to pH changes. Authors claimed the dynamic crosslinking of boronic ester bonds exhibits acid-reinforced mechanism. However, the reviewer cannot find any information about pH value in the main text and figure legend (Figure 2H). The same problem can be found in the Figure 3B which doesn’t reveal the pH value in the evaluation of tissue binding ability.

Response 1: Many thanks for your careful reviewing. It is our mistake to neglect the

key information about pH values in the main text and figure legend. According to the reviewer's suggestions, we have added the necessary pH value in the corresponding figure legend as "Fig. 2H Representative frequency sweep rheological plots of GMPD-*hν* and GMPD+*hν* hydrogels at pH 7.4. I The crossover frequencies (ω_c) obtained from frequency sweep rheology for GMPD-*hν* hydrogels at each pH value ranging from 4.5-8.5. J Final shear moduli of GMPD+*hν* hydrogels at each pH value ranging from 4.5-8.5." and "Fig. 3B Photographs of GMPD hydrogels (stained by fast green) formed *in situ* on muscle tissue and immersed in acidic solution (pH = 6.5)". The corresponding expression of "Importantly, the dynamic crosslinking of boronic ester bonds exhibited acid-reinforced mechanical strength with ω_c increasing as pH decreased (ranging from 4.5-8.5), which is very suitable for harsh UME of low pH values (Fig. 2H, I)." and "Fig. 3B depicted that there was no obvious breakage or detachment of the adhesive hydrogels regardless of stretching and twisting behavior upon acidic solution (pH = 4.5)." were also revised in the main text.

Comments 2: The research goal here is to provide a scarless urethral reconstruction for urethral injury. The proposed hydrogel should be used in the existing of urine. The pH range of urine could be in the range of 4.5-8 and is rich in urea. First, the only pH test in this manuscript I can find is Figure 2I and 2J which evaluated W_c and modulus in pH 6.5, 7.4, and 8.5. Authors should further check the material stability in lower pH conditions. Secondly, polycondensation is known to happen to boric acid and urea. May authors comment on this issue?

Response 2: Many thanks for the reviewer's insightful comments.

Regarding the physicochemical properties in acidic environment: According to the reviewer's suggestions, we have added the corresponding assessment of rheological properties and mechanical stability at pH 4.5. As shown in Fig. 2I, J and S5, the experimental results also showed the acid-reinforced mechanical strength and stability in the acidic urine environment. The corresponding expression of "Importantly, the dynamic crosslinking of boronic ester bonds exhibited acid-reinforced mechanical strength with ω_c increasing as pH decreased (ranging from

4.5-8.5), which is very suitable for harsh UME of low pH values (Fig. 2H, I). Moreover, time sweep rheological measurements demonstrated that secondary photopolymerization could effectively improve the elasticity of GMPD hydrogels without any influence of pH values (Fig. 2J).” and “As shown in Supplementary Fig. 5, GMPD-*hν* hydrogels in acid solution could effectively extend the *in vitro* degradation time from ~3 days (*pH* = 7.4) to ~14 days (*pH* = 4.5), which is attributed to the acid-reinforced structural stability of boronic ester bonds (correlated with Fig. 2I). Additionally, GMPD+*hν* hydrogels exhibited long-lasting stability over 14 days in any harsh UME (*pH* values from 4.5 to 7.4) due to the pH-independent covalent stabilization of photopolymerization.” were revised in the results section.

Fig. 2I The crossover frequencies (ω_c) obtained from frequency sweep rheology for GMPD-*hν* hydrogels at each pH value ranging from 4.5-8.5. J Final shear moduli of GMPD+*hν* hydrogels at each pH value ranging from 4.5-8.5.

Supplementary Fig. 5. The *in vitro* degradation curves of GMPD-*hν* and GMPD+*hν* hydrogels in neutral (*pH* = 7.4) or acidic solutions (*pH* = 6.5 and 4.5).

Regarding the potential side reaction: As the reviewer concerned, polycondensation side reaction would happen between boric acid and urea under high temperature condition in organic solvent as described “The precursor was synthesized by refluxing the mixture of boric acid and urea in xylene/toluene for about 4hrs.” [Reference: S. Mondal et. al. Polycondensation of Urea and Boric Acid to give Polyborate Ester, A Precursor for Boron Nitride, *Advanced Materials Research*, 29, 199-202 (2007)]. Thus, it is reasonable to infer that polycondensation side reaction is difficult to happen at room temperature in PBS solution between GMP and GMD polymers. In addition, it is hard for us to certify the possible side reaction of phenylboronic acid chemistry in the complex biomacromolecule system because the chemical crosslinking products between biomacromolecules can't be purified and analyzed by universal chemistry characterization.

Comments 3: The chemical composition of GMP and GMDs well as GMPD-hv and GMPD+hv should be provided. The Mw and Mn of the polymers and the ratio of FPBA and cis-diol group are important to the readers.

Response 3: Many thanks for the reviewer's professional suggestions. In fact, the biomacromolecules of gelatin have no definitive molecular weight due to the synthetic method by collagen hydrolysis, so it is impossible to supply the explicit Mw and Mn of the gelatin biomacromolecules.

According to the reviewer's suggestions, we further added the substitution degree of FPBA in GMP (~32%) and *cis*-diol group in GMD (~39%). The corresponding expression of “In our design, we first synthesized two types of gelatin-derived polymers: fluorophenylboronic acid (FPBA)-modified gelatin methacryloyl (GMP, ~32% FPBA-substituted degree; ~60% methacryloyl-substituted degree) and *cis*-diol-modified gelatin methacryloyl (GMD, ~39% diol-substituted degree; ~60% methacryloyl-substituted degree).” was revised in the results section. The corresponding methods of “Briefly, the methacryloyl-substituted degree was determined by the decrease of proton peak at 2.9 ppm (amino groups of gelatin); the FPBA-substituted degree was determined by the integral ratio of the proton peaks at

7.3-7.5 ppm to the peaks at 5.2-5.7 ppm (methacryloyl groups); the diol-substituted degree was determined by the integral ratio of the proton peaks at 3.1 ppm to the peaks at 5.2-5.7 ppm (methacryloyl groups).” were added in the methods section.

Comments 4: Boronic acid could be sensitive under photo-irradiation with the generation of phenyl radical intermediate. Authors may have to check if side reactions can be found.

Response 4: Many thanks for the reviewer’s professional suggestions. According to the reviewer’s suggestions, we have checked the possible radical intermediate by electron paramagnetic resonance (EPR) experiments. The current results certified that there was no phenyl radical intermediate under photo-irradiation as shown in the following figure. We speculate that it maybe attribute to the graft of phenylboronic acid groups onto the gelatin biomacromolecules that quench the generation of phenyl radical intermediate.

Supplementary Fig. The electron paramagnetic resonance characterization of GMP polymer with light irradiation (light: 365-nm LED, 20 mW/cm², 1 min irradiation time).

Comments 5: There are four XPS data including red, green, purple, and orange lines in each spectrum. Authors should provide details about what the color stands for in the figure legend.

Response 5: Many thanks for the reviewer’s professional suggestions. According to the reviewer’s suggestions, we have added the corresponding details of each spectrum in the figure legend of “Fig. 2D The C(1s) XPS regions of GMP, GMD, and

GMPD-*hν* hydrogel samples. Red line: C(1s) characteristic peak; green line: C=O (288.06 eV); orange line: C-O (286.37 eV); blue line: C-C (284.75 eV); blue arrows represent C(1s) regions. E The B(1s) XPS regions of GMP, GMD, and GMPD-*hν* hydrogel samples. Red line: B(1s) characteristic peak; blue line: B-O (191.34 eV); green line: B-C (189.75 eV); red arrows represent B(1s) regions.”

Reviewers' Comments:

Reviewer #1:

Remarks to the Author:

The Authors have answered most of my comments with the inclusion of additional experimental data. My remaining concerns are as follows:

1 The authors demonstrated that the GMPD+hv hydrogel could withstand the flow-induced shear force in vivo. However, since the hydrogel undergoes continuous flow-induced shear force, long-term adhesive performance should be considered. In vivo/Ex vivo adhesive capacity is suggested to be provided. This does not necessarily mean that the current manuscript is insufficient, but it is a limitation of the data as provided.

2 In Fig. 7E, the zoom-in H&E staining image of the GMPD-V/TI-treated rabbit urethral canal was improperly selected.

Reviewer #2:

Remarks to the Author:

The authors addressed properly the comments.

A couple of additional comments:

Additional pH dependent experiments were only performed at 4.5. In the figures, the pH jumps from pH 4.5 to 6.5. The readers may find this confusing, particularly because pH values of 5-5.5 are the most commonly found in urine. Please, include the pH 5.5 in those set of experiments.

Histological images with H&E w (Fig 7) with much higher magnification are needed, specially focusing on the highlighted areas (in red) of the urothelium.

Reviewer #3:

Remarks to the Author:

The authors tried to answer most questions raised by the reviewer. The current vision is suitable for publication.

Reviewer 1 comments and our corresponding answers:

Overall comments:

The Authors have answered most of my comments with the inclusion of additional experimental data. My remaining concerns are as follows:

Response:

Thanks for the reviewer's positive comments and professional suggestions on our work. According to the professional suggestions, we had tried our best to revise the manuscript and our responses point by point according to the reviewer's comments were listed as bellows.

Comments 1:

1 The authors demonstrated that the GMPD+hv hydrogel could withstand the flow-induced shear force *in vivo*. However, since the hydrogel undergoes continuous flow-induced shear force, long-term adhesive performance should be considered. *In vivo/Ex vivo* adhesive capacity is suggested to be provided. This does not necessarily mean that the current manuscript is insufficient, but it is a limitation of the data as provided.

Response:

Many thanks for the reviewer's insightful comment and professional suggestion. As the reviewer's concern, the long-term adhesive performance is important to withstand the flow-induced shear force *in vivo*.

It needs to be pointed that we could only provide the *ex vivo* adhesive capacity after 12 hours because the hydrogels would be dried upon long-term exposing to air. Thus, it is difficult to provide the long-term *ex vivo* adhesive capacity. The corresponding expression of "The *ex vivo* adhesive performance was basically unchanged after 12 hours, which verified the stability of hydrogel-tissue adhesion (Supplementary Fig. 4)." was added in the main text.

Supplementary Fig. 4. The statistical data of standard lap shear (A) and incision sealing (B) tests showing the *ex vivo* adhesive performance after 0 and 12 hours.

In addition, the long-term quantitative data of *in vivo* adhesive capacity was also hardly evaluated due to the gradual degradation tendency of the GMPD hydrogels on tissue surface. Instead, we could qualitatively observe the remaining hydrogels on the tissue surface, which indirectly showed the satisfactory long-term adhesive performance *in vivo*.

Supplementary Fig. 7. A, B) Representative photographs (A) and histological examinations of H&E and Masson's trichrome staining (B) of the remaining **GMPD** hydrogels in the urethral defect model after 3-, 7-, 14-days surgery. Blue circles and red arrows represent the remaining hydrogels.

Comments 2:

2 In Fig. 7E, the zoom-in H&E staining image of the GMPD-V/TI-treated rabbit urethral canal was improperly selected.

Response:

Many thanks for your careful reviewing. According to the reviewer's suggestions, we have revised the zoom-in H&E staining image of the GMPD-V/TI-treated rabbit urethral canal in Fig. 7E.

Fig. 7 *In vivo* scarless urethral reconstruction using GMPD hydrogel dressings in rabbits. **A** Photographs of the surgical operation of *in situ* urethral defect repair using GMPD hydrogel dressings. **B, C** Urethrography images (**B**) and the corresponding blockage ratios (**C**) of the rabbit urethral canal in the GMPD, GMPD-V, GMPD-TI, GMPD-V/TI, and control (Ctrl) groups after 4- and 8-weeks surgery. Red arrows represent urethral stricture. Blue arrows represent urethral patency. **D, E** Gross view (**D**) and histological examinations of H&E and Masson's trichrome staining (**E**) of the rabbit urethral canal in the GMPD, GMPD-V, GMPD-TI, GMPD-V/TI, and control (Ctrl) groups after 8-weeks surgery. GMPD hydrogels: 10% w/v, GMP: GMD = 1:1; GMPD-V hydrogels: 10% w/v GMPD with 1% w/v V-GM; GMPD-TI hydrogels: 10% w/v GMPD with 1% w/v TI-PLGA; GMPD-V/TI hydrogels: 10% w/v GMPD with 1% w/v V-GM and 1% w/v TI-PLGA; light: 365-nm LED, 20 mW/cm², 30 s irradiation time. (n = 4, **** p < 0.0001)

Reviewer 2 comments and our corresponding answers:

Overall comments:

The authors addressed properly the comments.

A couple of additional comments:

Response:

Thanks for the reviewer's positive comments and professional suggestions on our work. According to the professional suggestions, we had tried our best to revise the manuscript and our responses point by point according to the reviewer's comments were listed as bellows.

Comments 1:

Additional pH dependent experiments were only performed at 4.5. In the figures, the pH jumps from pH 4.5 to 6.5. The readers may find this confusing, particularly because pH values of 5-5.5 are the most commonly found in urine. Please, include the pH 5.5 in those set of experiments.

Response:

Many thanks for the reviewer's professional suggestion. According to the reviewer's suggestion, we have supplemented the corresponding data (such as frequency sweep rheology, swelling ratio, and degradation rate) at pH value of 5.5.

Fig. 2 Physicochemical characterization of UME-adaptable GMPD hydrogels. **A** Schematic of hybrid crosslinking mechanisms obtained by combining boronic ester dynamic crosslinking (viscous segment) and photopolymerization (elastic segment) to construct viscoelastic **GMPD** hydrogels. **B** Photographs show gelation steps through two-component mixing followed by light irradiation. **C** ¹H NMR spectra evolution of **GMPD** hydrogels with or without light irradiation. Grey arrows represent the proton peak of methacryloyl groups. **D** The C(1s) XPS regions of **GMP**, **GMD**, and **GMPD-hv** hydrogel samples. Red line: C(1s) characteristic speak; green line: C=O (288.06 eV); orange line: C-O (286.37 eV); blue line: C-C (284.75 eV); blue arrows represent C(1s) regions. **E** The B(1s) XPS regions of **GMP**, **GMD**, and **GMPD-hv** hydrogel samples. Red line: B(1s) characteristic speak; blue line: B-O (191.34 eV); green line: B-C (189.75 eV); red arrows represent B(1s) regions. **F** ATR-FTIR spectra of **GMPD** hydrogels with or without light irradiation. **G** Representative time sweep rheological plots of **GMPD-hv** hydrogels. **H** Representative frequency sweep rheological plots of **GMPD-hv** and **GMPD+hv** hydrogels at pH 7.4. **I** The crossover frequencies (ω_c) obtained from frequency sweep rheology for **GMPD-hv** hydrogels at each pH value ranging from 4.5-8.5. **J** Final shear moduli of **GMPD-hv** hydrogels at each pH value ranging from 4.5-8.5. **GMPD** hydrogels: 10% w/v, **GMP**: **GMD** = 1:1; **GMPD-hv** hydrogels: **GMPD** hydrogels without light irradiation; **GMPD+hv** hydrogels: **GMPD** hydrogels with light irradiation; light: 365-nm LED, 20 mW/cm², 30 s irradiation time.

Supplementary Fig. 6. The *in vitro* degradation curves of **GMPD-hv** and **GMPD+hv** hydrogels in neutral (pH = 7.4) or acidic solutions (pH = 4.5-6.5).

Supplementary Fig. 8. The *in vitro* swelling ratio of **GMPD+hv** hydrogels in different pH values ranging from 4.5-8.5.

Comments 2:

Histological images with H&E (Fig 7) with much higher magnification are needed, specially focusing on the highlighted areas (in red) of the urothelium.

Response:

Many thanks for your careful reviewing. According to the reviewer's suggestions, we have supplemented the corresponding magnified images of H&E histological

staining in the Supplementary Fig. 14.

Supplementary Fig. 14. The magnified H&E staining of the rabbit urethral canal in the **GMPD**, **GMPD-V**, **GMPD-TI**, **GMPD-V/TI**, and control (**Ctrl**) groups after 8-weeks surgery.

Reviewer 3 comments and our corresponding answers:

Overall comments:

The authors tried to answer most questions raised by the reviewer. The current version is suitable for publication.

Response:

Thanks for the reviewer's positive affirmation on our work.

Reviewers' Comments:

Reviewer #1:

Remarks to the Author:

The authors have addressed related issues.

Reviewer #2:

Remarks to the Author:

The authors addressed all concerns